# Precipitation over southern Africa: Is there consensus among GCMs, RCMs and observational data?

Maria Chara Karypidou[1], Eleni Katragkou[1], Stefan Pieter Sobolowski[2]

[1] Department of Meteorology and Climatology, School of Geology, Faculty of Sciences, Aristotle University of Thessaloniki, Thessaloniki, Greece
[2] NORCE Norwegian Research Centre, Bjerknes Centre for Climate Research, Bergen, Norway

*Correspondence to*: karypidou@geo.auth.gr

**Abstract.** The region of southern Africa (SAF) is highly vulnerable to the impacts of climate change and is projected to experience severe precipitation shortages in the coming decades. Ensuring that our modelling tools are fit for the purpose of assessing these changes is critical. In this work we compare a range of satellite products along with gauge-based datasets. Additionally, we investigate the behaviour of regional climate simulations from the Coordinated Regional Climate Downscaling Experiment (CORDEX) – Africa domain, along with simulations from the Coupled Model Intercomparison Project Phase 5 (CMIP5) and Phase 6 (CMIP6). We identify considerable variability in the standard deviation of precipitation between satellite products that merge with rain gauges and satellite products that do not, during the rainy season (Oct-Mar), indicating high observational uncertainty for specific regions over SAF. Good agreement both in spatial pattern and the strength of the calculated trends is found between satellite and gauge-based products, however. Both CORDEX-Africa and CMIP5 ensembles underestimate the observed trends during the analysis period. The CMIP6 ensemble displayed persistent drying trends, in direct contrast to the observations. The regional ensembles exhibited improved performance compared to their forcing (CMIP5), when the annual cycle and the extreme precipitation indices were examined, confirming the added value of the higher resolution regional climate simulations. The CMIP6 ensemble displayed a similar behaviour to CMIP5 but reducing slightly the ensemble spread. However, we show that reproduction of some key SAF phenomena, like the Angolan Low (which exerts a strong influence on regional precipitation), still poses a challenge for the global and regional models. This is likely a result of the complex climatic process that take place. Improvements in observational networks (both in-situ and satellite), as well as continued advancements in high-resolution modelling will be critical, in order to develop a robust assessment of climate change for southern Africa.

## 1 Introduction

The region of Sub-Saharan Africa has been characterized as one of the most vulnerable regions to climate change (Kula et al., 2013; Serdeczny et al., 2017) and more specifically, the region of southern Africa (SAF) has been identified as a climate change hotspot (Diffenbaugh and Giorgi, 2012). Taking into consideration that the majority of the population living in SAF (70%) is dependent on rainfed agriculture (Mabhaudhi et al., 2018), any climate change induced alteration of the spatiotemporal patterns of precipitation will require a rapid adaptation of the agricultural sector. Concurrently, SAF is also

characterized by low adaptive capacity to changes in climatic conditions (Davis and Vincent, 2017), hence, it emerges as a high risk region. In addition, approximately 26% of the SAF population is undernourished (AFDB, 2019). This figure is expected to increase significantly by 2050 (Tirado et al., 2015). Apart from the impacts on the agricultural sector though, climatic changes are expected to alter the spatiotemporal patterns of vector-borne disease occurrence (Rocklöv and Dubrow, 2020), cause severe damage to infrastructure and road networks (Chinowsky et al., 2015) and exacerbate poverty (Azzarri

and Signorelli, 2020). Due to these impacts it is critical that the current spatiotemporal patterns of precipitation are accurately reproduced by our modelling systems and observations (whether in-situ, reanalysis or satellite) over SAF. Only then can we credibly assess future climate change impacts and inform strategies aiming to mitigate their effects on local communities.

Towards this end, satellite, gauge-based and reanalysis products are extensively used, in order to monitor current spatial

and temporal precipitation patterns and to further characterize precipitation variability and change during recent decades. For future projections however, climate models able to simulate the (thermo)dynamical processes of the atmosphere are employed. Such an endeavor has been performed in the context of the Coupled Model Intercomparison Project Phase 5 (CMIP5) (Taylor et al., 2012) using General Circulation Models (GCMs) and in the context of the Coordinated Regional Climate Downscaling Experiment (CORDEX) – Africa domain (Giorgi and Gutowski, 2015) using Regional Climate

Models (RCMs). The latest advancement in the climate modelling community involves GCMs and earth system models (ESMs), participating in the CMIP6 ensemble, providing input for the 6[th] Assessment Report of the Intergovernmental Panel on Climate Change (IPCC) (Eyring et al., 2016). However, the confidence with which one can claim future climate projections produced by GCMs, ESMs or RCMs are fit for purpose, is usually assessed based on their ability to simulate current climatic conditions. For instance, Munday and Washington (2018) showed that the CMIP5 ensemble displayed a

systematic wet bias over the SAF region that was caused by the misrepresentation of orographic features located over the area of Tanzania. A wet bias caused by structural model errors was also identified in the dynamically downscaled and higher-resolution CORDEX-Africa ensemble (Kim et al., 2014). Therefore, a valid question arises as to what the most suitable dataset is, with which climate impact studies can be fed with when the SAF region is concerned. In addition, before the task of characterizing future precipitation trends is addressed, it is imperative to diagnose the degree to which observed

precipitation trends over the recent decades are reproduced by GCMs and RCMs.

A comprehensive analysis of the performance of the CORDEX-Africa ensemble over Africa was first presented in Nikulin et al. (2012). They showed that during the rainy season (Jan-Mar as used in Nikulin et al. (2012)) there is a weak wet bias over southern Africa, and that the use of the ensemble mean was able to outperform individual models, highlighting the importance of ensemble-based approaches. The Nikulin et al. (2012) analysis was conducted on a pan-African scale.

Similarly, Kalognomou et al. (2013) analyzed the same ensemble of CORDEX-Africa simulations, focusing over southern Africa and reported similar findings. In Meque and Abiodun (2015) the same ensemble of 10 evaluation simulations was again used, but it was also compared with a set of CMIP5 GCM simulations, with the purpose to identify a causal association between ENSO and drought events over southern Africa. In Meque and Abiodun (2015) it was stated that RCMs were able to provide added value, compared to their driving GCMs. A comprehensive assessment of the added value between historical

CORDEX-Africa RCMs simulations and of their driving CMIP5 GCMs on a seasonal timescale over the whole of Africa, was performed in Dosio et al. (2019). The first time the CORDEX-Africa ensemble is compared to both CMIP5 and CMIP6 ensembles is presented in Dosio et al. (2021). More specifically, in Dosio et al. (2021) the analysis is performed on a seasonal timestep and on pan-African scale and its particular emphasis is placed on the projected changes of future precipitation, although a part of the analysis is dedicated to the period 1981-2010.

Satellite and gauge-based datasets display increasing trends during the historical period (Gu et al., 2016; Harrison et al., 2019) for annual and seasonal precipitation over SAF (32-42 mm year[-1] per decade), an observation that is also identifiable in the Atmospheric Model Intercomparison Project (AMIP), but not in CMIP5 (Maidment et al., 2015). During DJF, precipitation trends over SAF display a remarkably robust signal in gauge-based, satellite and AMIP datasets (Maidment et al., 2015). In addition, Onyutha (2018) also reported on the increasing precipitation trends over SAF during DJF, especially

after the 1960's. However, according to CMIP5, precipitation is projected to decrease over SAF in the 21[st] century (IPCC, 2013). This estimate also holds for simulations performed using RCMs forced with CMIP5 (Pinto et al., 2016; Dosio et al., 2019b). The increase of the observed precipitation trends over SAF has been attributed to the recent strengthening of the Pacific Walker Circulation (Maidment et al., 2015), which is captured in observational datasets and in AMIP simulations, but not in CMIP5 (L'Heureux et al., 2013; Yim et al., 2016). CMIP6 displays an even more robust future decline in

precipitation and increase of drought events over SAF, relative to its predecessor (Ukkola et al., 2020). However, although the CMIP6 ensemble exhibits multiple improvements on various levels (Wyser et al., 2020), certain biases and challenges identified in CMIP5 during the historical period persist in CMIP6 (Kim et al., 2020).

RCMs are known to add value to climate simulations over regional scales, mainly because the spatial resolution increases, resolving atmospheric waves in a more detailed manner and also, because surface characteristics interacting with the

atmosphere are represented more accurately (Denis et al., 2003; Giorgi et al., 2014). Considering the aforementioned challenges displayed in the CMIP5 simulations to accurately capture precipitation amounts under current climatic conditions and recent precipitation trends, we investigate the degree to which this observation holds also for RCMs, forced with GCMs participating in the CMIP5 ensemble. Theory tells us that RCMs develop their own physics. However, often times the impact

of the driving GCMs is evident on the RCM simulations (Denis et al., 2003; Laprise et al., 2008; Di Luca et al., 2013; Giorgi, 2019).

Therefore, in this paper we expand on previous research to investigate how monthly precipitation during the rainy season over southern Africa is simulated by different modelling systems, by analyzing the monthly precipitation climatologies, the interannual variability, specific precipitation indices and monthly precipitation trends during the period 1986-2005, in four different modeling systems (CORDEX 0.22º/0.44º, CMIP5/6) and observational ensembles (satellite, reanalysis and gridded datasets). Our main goal is to provide a comprehensive overview with regards to precipitation climatology over SAF as simulated by the state-of-the-art tools used by climate scientists. In addition, we investigate whether higher resolution models are able to provide an improved representation of precipitation over southern Africa and we investigated how a particularly important atmospheric feature, the Angola Low (AL) pressure system, is simulated in the RCM and GCM ensembles.

In Section 2 the data used are presented along with the methodology employed. In Section 3 the results are presented. More specifically, the results are analyzed based on the monthly climatology, the annual cycle of precipitation, the AL pressure system, the ETCCDI precipitation indices and the monthly precipitation trends. Lastly, in Section 4 we provide the discussion of the analysis along with some concluding remarks.

## 2 Data and methodology

### 2.1 Data

We analyse daily and monthly precipitation from 5 types of datasets, namely observational datasets (OBS), GCMs and ESMs that comprise the CMIP5 and CMIP6 ensembles and regional climate models (RCMs) that comprise the CORDEX-Africa ensemble at 0.44º of spatial resolution (CORDEX0.44) and at 0.22º of spatial resolution (CORDEX0.22). The analysis is concerned with the SAF region, which is defined as the area between 10 ºE to 42 ºE and 10 ºS to 35 ºS. The analysed period is 1986-2005, as this is the period during which the estimates of all 5 aforementioned datasets overlap. Although satellite and reanalysis products cannot be termed as purely "observational", in the context of the current work they are classified as such, in order to differentiate them from climate model datasets (CORDEX0.44, CORDEX0.22, CMIP5, CMIP6). Hereafter "OBS" refers to satellite, gauge-based and reanalysis products.

### 2.1.1 Observational data

The OBS data used are based on the analysis of Le Coz and van de Giesen (2020) and are comprised of 5 gauge-based products (datasets that are derived by spatial interpolation of rain gauges and station data: CRU.v4.01, UDEL.v7,

PREC/L.v0.5, GPCC.v7, CPC-Global.v1), 6 satellite products (given below) and 1 reanalysis product, ERA5. The datasets have a temporal coverage that extends through the analysed period (1986-2005). The gauge-based products were chosen so

that they have a spatial resolution less than or equal to $0.5^{\circ}$ x $0.5^{\circ}$ and the satellite products have a spatial resolution less or equal to $0.25^{\circ}$ x $0.25^{\circ}$. For satellite products however, there was an exception for 2 products (CMAP.v19.11 and GPCP.v2.2) with a resolution equal to $2.5^{\circ}$ x $2.5^{\circ}$ that were also included in the analysis due to their wide use in the literature. The OBS ensemble is made of 12 products. More details concerning the OBS datasets are provided in Table S1. In certain parts of the following analysis the OBS products are either used collectively or they are split into sub-ensembles, based on the method(s)

used for their production. More specifically, these sub-ensembles are the mean of all gauge-based precipitation products (Gauge-Based), the ensemble mean of satellite products that merge with rain gauges (Satellite-Merge) (ARC.v2, CMAP.v19.11, GPCP.v2.2) and the ensemble mean of satellite products that do not merge directly with rain gauges (Satellite-NoMerge) (CHIRPS.v2, TAMSAT.v3, PERSIANN-CDR), but they use alternative methods such as calibration, bias adjustment or artificial neural network techniques (Le Coz and van de Giesen, 2020).

**2.1.2 Climate model simulations**

We retrieved daily precipitation for a set of 26 RCM simulations performed as part of CORDEX-Africa historical simulations at $0.44^{\circ}$ (~50 km) spatial resolution, comprising the CORDEX0.44 ensemble. We also retrieved a set of 10 RCM simulations performed also within CORDEX-Africa, as part of the CORDEX-CORE project (Coppola et al., 2021), available at $0.22^{\circ}$ (~25 km) spatial resolution (CORDEX0.22). In addition, daily precipitation was retrieved for a set of 10 CMIP5

GCMs, with 3 additional simulations with variations in the GCM's resolution (IPSL-LR/IPSL-MR), the ocean model (GFDL-ESM2M/GFDL-ESM2G) and Realization/Initialization/Physics (ICHCE-EC-EARTH-r1i1p1/ ICHCE-EC-EARTH-r12i1p1). The CMIP5 models selected were the ones used as forcing in the CORDEX0.44 historical simulations. In total, precipitation from a set of 13 CMIP5 simulations was used. Additionally, we exploited daily precipitation from a set of 8 CMIP6 GCM and ESM simulations. The CMIP6 simulations selected were performed with the updated versions of the same

models that were part of the CMIP5 ensemble. This selection served to construct CMIP5 and CMIP6 ensembles that were comparable. Precipitation data for all simulations were retrieved from the Earth System Grid Federation (ESGF). In addition, we retrieved temperature at 850 hPa for both CORDEX0.44/0.22 from ESGF. For the CMIP5 and CMIP6 simulations temperature and geopotential height at 850 hPa was retrieved from the Climate Data Store (CDS). Geopotential height at 850 hPa was not available for CORDEX-Africa simulations. Lastly, elevation data for CORDEX-Africa and CMIP5 were

obtained from ESGF, while the Shuttle Radar Topography Mission (SRTM) (Farr et al., 2007) Digital Elevation Model was used as the observed elevation in the topography transects for a selected latitude over SAF. Details about the models used are provided in Tables S2-S5.

**2.2 Methodology**

Precipitation climatologies are investigated on a monthly basis, due to the fact that precipitation over SAF arises as the result of atmospheric mechanisms that display high variability during the rainy season. The aggregation of precipitation to seasonal means might often obscure certain spatial characteristics that are better identified on a monthly basis. The within-ensemble agreement is investigated using the sample standard deviation (SD), which is calculated using monthly mean values over the period 1986-2005 for each model (or observational dataset) separately. We also employ 4 precipitation

indices constructed in the context of the Expert Team on Climate Change Detection and Indices (ETCCDI) (Peterson and Manton, 2008), utilising daily precipitation amounts for the period 1986-2005. The 4 ETCCDI indices are used to describe total annual precipitation (PRCPTOT), annual maximum daily precipitation (Rx1Day), annual number of days with daily precipitation >10 mm (R10mm) and annual number of days with daily precipitation >20 mm (R20mm). These indices are calculated for each individual simulation (CMIP5, CMIP6, CORDEX0.44 and CORDEX0.22), and OBS products,

separately and yield a value for every year (Jan-Dec) during the period 1986-2005. The calculation of indices required data having a daily temporal resolution, hence, observational datasets that provided monthly aggregates are excluded. The spatial averages calculated over SAF for the annual cycle and the ETCCDI indices consider land pixels only. For the construction of ensemble means, either in observational or model ensembles, datasets were remapped to the coarser grid using conservative remapping for precipitation and bilinear interpolation for temperature at 850 hPa.

In order to investigate some basic thermodynamical aspects that may differentiate precipitation in the CMIP5/6 and the CORDEX0.44/0.22 ensembles, we look into the seasonal representation of the Angola Low (AL) pressure system over SAF. The AL pressure system is a semi-permanent synoptic scale system, that plays a strong role in modulating precipitation over SAF (Reason and Jagadheesha, 2005; Lyon and Mason, 2007; Crétat et al., 2019; Munday and Washington, 2017; Howard and Washington, 2018). More specifically, the reason why we chose to put an emphasis on the AL pressure system, is that

the AL redistributes low-tropospheric moisture entering SAF from the southern Atlantic and the southern Indian oceans and also, moisture transport originating from the Congo basin. In addition, AL events precede the formation of Tropical Temperate Troughs (TTTs) and hence, they can be considered as their precursor in the "climate process chain (Daron et al., 2019). As stated in Howard and Washington (2018), it is common that AL events precede TTT events, since the AL pressure system functions as a key process necessary for the transport of water vapor from the tropics towards the extratropics (Hart

et al., 2010).

In Munday and Washington (2017) AL events were identified using geopotential height at 850 hPa. However, since geopotential height is not available for CORDEX0.44/0.22 simulations, we could not employ this method. Hence, based on the variables that are already available within both CORDEX and CMIP5/6 ensembles, we use potential temperature at 850 hPa (theta850) as an alternative "proxy" variable that provides thermodynamical information. In order to ensure that

theta850 can be used instead of zg850, we examine the relationship between theta850 and zg850 over the study region in

ERA5, for each month of the rainy season (Oct-Mar), using the climatological mean monthly values for the period 1986-2005 (Fig. S1, S2). As shown in Fig. S1, during October over the south-eastern part of Angola, there is a region of low pressures. Moving towards the core of the rainy season, the low-pressure system deepens, while there seems to be a weak extension of low pressures towards the south. Also, as shown in Fig. S2, during October there is an area of high theta850 values located over south-eastern Angola, coinciding with the region of low zg850 values. As stated in Munday and Washington (2017), this is indicative of the dry convection processes that are at play during the beginning of the rainy season over the region. Moving towards DJF, the high theta850 values move southwards, indicating that in the core of the rainy season, convection over the greater Angola region is not thermally induced, but there is a rather dynamical large-scale driver. In Fig. S3 the scatterplots between zg850 (x-axis) and theta850 (y-axis) for each month of the rainy season are shown, over the whole southern Africa (land pixels only). The same plot, but with pixels only from the greater Angola region (14 °E to 25 °E and from 11 °S to 19 °S) is displayed in Fig. S4. Although the relationship between the two variables is not linear, they display a considerable association, especially over the greater Angola region.

In Howard and Washington (2018) AL events are identified using daily relative vorticity ($\zeta$) at 800 hPa. Since u and v wind components are not available at 800 hPa (but at 850 hPa) for the CORDEX ensembles, we investigate whether the 850 hPa pressure level can be used instead. We also examine whether the $\zeta$ threshold has to be adjusted. In Howard and Washington (2018), AL events are identified within the region ranging from 14 °E to 25 °E and from 11 °S to 19 °S for mean daily $\zeta$ values $<-4 \times 10^{-5}$ s$^{-1}$. An additional issue that we take into account, is that u and v wind components are not available on a daily timestep for CMIP6, but only on a monthly timestep. Hence, for consistency reasons we work with monthly files in all ensembles (both CMIP, CORDEX) and in ERA5.

With regards to the question of whether the 850 hPa pressure level can be used instead of 800 hPa, we examine monthly relative vorticity in ERA5 in both pressure levels, within the region from 14 °E to 25 °E and from 11 °S to 19 °S (Fig. S5). Both distributions are very similar in shape, maxima and spread, although the distribution of $\zeta$ values at 800 hPa appear to have a shorter tail. On both panels, both the Howard and Washington (2018) and the Desbiolles et al. (2020) thresholds are indicated. We conclude that the 850 hPa pressure level can be used instead of 800 hPa. With regards to the fact that u and v wind components are available only on a monthly timestep in CMIP6, we compare the daily and monthly relative vorticity values at 800 hPa in ERA5 for all the months of the rainy season (Oct-Mar) (Fig. S6). The difference in the y-axis results from the fact that when $\zeta$ is calculated using a daily timestep, the histogram is drawn using 5.421.825 values, while when the $\zeta$ is calculated using monthly u and v values, it is drawn using 178.200 values (for the period 1986-2005). As shown, the distribution of the monthly values has a much shorter tail and the Howard and Washington (2018) threshold appears to be very strict, as a criterion for the identification of AL events.

Concerning the question of what the optimal threshold for the identification of AL events in all datasets is, we investigate the statistical distribution of mean monthly cyclonic vorticities in all ensembles used, for the 850 hPa pressure level (Fig. S7). We conclude that the threshold used in Desbiolles et al. (2020) ($\zeta$ values $<-1.5 \times 10^{-5}$ s$^{-1}$) is reasonable, considering the shape of the distributions examined. However, when the Desbiolles et al. (2020) threshold is applied to the data, it is also

found to be too strict, especially for CMIP5/6. Hence, we identify AL events having $\zeta$ <-0.00001 s$^{-1}$. Lastly, we use geopotential height at 850 hPa for visual inspection only in ERA5 and CMIP5/6 ensembles.Lastly, the Theil-Sen's slope (Theil, 1992; Sen, 1968) for monthly precipitation during the period 1986-2005 is calculated for each dataset. This is a non-parametric approach to estimate trends, that is insensitive to outliers. Statistical significance is assessed using the Mann-Kendall test (Mann, 1945; Kendall, 1948).

**3 Results**

**3.1 Climatology**

Figure 1 displays monthly precipitation climatologies during Oct-Mar (rainy season over the study region) for ERA5 and for the ensemble means of 7 additional types of datasets. At the beginning of the rainy season (Oct) all products display precipitation maxima at the north-western part of the study region. Another precipitation maxima is observed at eastern
South Africa. For both regions, there is a slight tendency for gauge-based products to yield approximately 1 mm d$^{-1}$ less precipitation than reanalysis and satellite products. The CMIP5, CMIP6, CORDEX0.44 and CORDEX0.22 ensembles are also in agreement with regards to the location and amounts, however, CORDEX0.44 displays approximately 2 mm d$^{-1}$ more precipitation over Angola. During November, the rainband extends southwards and the region over South Africa experiencing high precipitation enlarges.

Moving towards the core of the rainy season (DJF) the precipitation maxima extends southwards following the collapse of the Congo air boundary (CAB) (Howard and Washington, 2019) and high precipitation amounts are also observed over the eastern part of the study region. More specifically during January, high precipitation amounts (>10 mm d$^{-1}$) are observed over an extended region in northern Mozambique for non-merging satellite products (Satellite-NoMerge). This area is also identified as a region of high precipitation in gauge-based products and in merging satellite products, however, with a
smaller magnitude. In ERA5, the spatial pattern of precipitation is more patchy and exhibits higher than observed precipitation amounts in the wider region of lake Malawi, reaching extremely high values (34 mm d$^{-1}$), as also indicated in the known precipitation issues of ERA5 over Africa (Hersbach et al., 2020). During DJF both CORDEX0.44 and CORDEX0.22 ensembles display precipitation values >3 mm d$^{-1}$ over almost all of the SAF region. This observation is also consistent in CMIP5 and CMIP6, however, maximum precipitation amounts in CMIP5 and CMIP6 are approximately >3
mm d$^{-1}$ larger than in the CORDEX ensembles. It is noteworthy, that in CORDEX0.22 during DJF, there are parts over northern SAF experiencing precipitation amounts >10 mm d$^{-1}$, a feature that is not seen in any of the observational products. After investigating the individual ensemble members used in the CORDEX0.22 ensemble (Fig. S8), we see that the excess amount of precipitation is removed from the CORDEX0.22 ensemble mean when RegCM4-7 simulations are not included (Fig. S9). In March, the rainband starts its northward shift, nevertheless, high precipitation amounts are still observed over

the eastern parts of the study region and over the coastal region of Angola. The retreat of the rainband is evident in both CORDEX0.44 and CORDEX0.22 however, CMIP5 and CMIP6 still exhibit extended regions of high precipitation.

     In Fig. 2, SD values for the 7 ensembles are presented during months Oct-Mar for the period 1986-2005 expressed as mm d$^{-1}$. SD is used as a measure of the within-ensemble agreement. As it is shown for gauge-based products, during October and November high SD values are observed primarily over Angola. For months Dec-Mar Angola remains a high SD region,
however, increased SD values are also observed over the eastern parts of SAF and especially over northern Mozambique. An important aspect influencing gauge-based products is the spatiotemporal coverage of the rain gauges used (Le Coz and van de Giesen, 2020), which is highly variable between regions and reporting periods. More specifically, after the 1970's the rain gauge coverage over Africa has decreased significantly (Janowiak, 1988) and the gauge network has been particularly sparse over the SAF region (Lorenz and Kunstmann, 2012; Giesen et al., 2014), which further implies that gauge-based products
depend on extrapolating values from surrounding gauges. Therefore, station density and the interpolation method employed are key factors in determining the accuracy of the final product (Le Coz and van de Giesen, 2020). The high SD values over Angola, are mainly due to the scarcity of available rain gauges used in the interpolation method (Fig. S10). After 1995, there is a noticeable reduction of the station/rain gauge data used over the SAF region (Fig. S11) for 3 of the gauge-based products.

A similar spatiotemporal pattern of SD is also observed in satellite-based products (Sat-Merge) which employ algorithms that merge rain gauges with thermal-infrared (TIR) images. This is indicative of the strong impact that the location and number of rain gauges exert on satellite algorithms that employ merging techniques (Maidment et al., 2014, 2015). The spatiotemporal pattern of SD for satellite-based products that do not merge with gauges (Sat-NoMerge) displays low SD values for October and November, however, during DJF localized areas of high SD appear over Angola, Zambia, Malawi
and Mozambique. The satellite products used in this ensemble are based on TIR images and precipitation is indirectly assessed through cloud top temperature (Tarnavsky et al., 2014; Ashouri et al., 2015; Funk et al., 2015). Hence, the occurrence and severity of precipitation is calculated based on a temperature threshold. In cases that the threshold is set to very low cloud top temperature values, the algorithm has high skill at identifying deep convection, however, warm rain events are not adequately captured (Toté et al., 2015). As it is shown in Fig. 2, high SD values in non-merging satellite
products are primarily observed over coastal regions and over regions where the elevation increases rapidly. These type of regions can be associated with orographic or frontal lifting of air masses (Houze, 2012), resulting in precipitation, without the threshold temperature of the cloud top being reached.

     In the CORDEX0.44 ensemble SD values are >0.8 mm d$^{-1}$ over almost all of the SAF region, however, very high SD values (3-9.8 mm d$^{-1}$) are observed in the coastal part of Angola and over the lake Malawi region during Nov-Mar. SD
values in CORDEX0.22 are considerably larger throughout the greater part of SAF, especially during DJF. In the CMIP5 ensemble the spatiotemporal pattern of SD values exceeds 2 mm d$^{-1}$ during Nov-Mar throughout the whole SAF region. CMIP6 displays a similar SD pattern. During March however, CMIP6 displays a substantial improvement in the agreement between its ensemble members. Overall, for the whole extent of SAF, the CORDEX-Africa ensembles display greater

agreement among ensemble members, however SD values become large over specific localized regions, mainly at western Angola and in the Malawi region. The CMIP5 and CMIP6 ensembles although not displaying the localized extreme SD values as CORDEX-Africa, displays generally high SD values throughout the whole extent of SAF.

## 3.2 Annual cycle

Figure 3 displays the annual cycle of precipitation in theCORDEX0.44, CORDEX0.22, CMIP5, CMIP6 and observational ensembles for land grid points. All datasets capture the unimodal distribution of precipitation over SAF, however considerable differences in precipitation amount and spread are observed.

Specifically, the CMIP5 ensemble exhibits significantly higher precipitation amounts than both CORDEX and observational ensembles. This difference becomes particularly pronounced during the rainy season, with CMIP5 yielding approximately 2 mm d$^{-1}$ more precipitation than the observational ensemble. It is also notable that for Nov-Feb, even the driest ensemble members of CMIP5 yield approximately 1 mm d$^{-1}$ more precipitation than the wettest ensemble members of the observational data. This is in agreement with Munday and Washington (2018) who identified a systematic wet bias over SAF in CMIP5, that was associated with an intensified north-easterly transport of moisture that erroneously reaches SAF, due to the poorly represented orography in the region of Tanzania and Malawi (which would hinder moisture originating from the Indian ocean from reaching SAF and instead force it to recurve towards the region of Madagascar). The behaviour of CMIP6 is similar to CMIP5, with a slightly smaller ensemble spread during Jan-Mar and a considerable reduction in spread during November.

The CORDEX0.44 ensemble reduces precipitation amounts during the core of the rainy season (DJF) compared to CMIP5, however, its behavior during the rest of the months is complicated. More specifically, during Aug-Oct CORDEX0.44 displays slightly higher precipitation amounts compared to CMIP5. During November, the difference between the CORDEX0.44 and the CMIP5 ensembles becomes noticeable, with the CMIP5 ensemble mean becoming 0.4 mm d$^{-1}$ larger than the CORDEX0.44 ensemble mean. During DJF the differences between the 2 ensembles maximize, with the CORDEX0.44 ensemble displaying good agreement with the OBS ensemble (<1 mm d$^{-1}$ difference in the ensemble means of CORDEX0.44 and OBS). From March until July, the difference between the CORDEX0.44 and CMIP5 ensembles starts to reduce gradually. The ensemble mean of the CORDEX0.22 ensemble is similar to that of the CORDEX0.44 ensemble, however its spread during the rainy season is considerably larger. Taking into consideration that excess precipitation in the CORDEX0.22 ensemble is introduced by RegCM4-7, we observe that the ensemble spread of the CORDEX0.22 ensemble is reduced, when RegCM4-7 is not included in the CORDEX0.22 ensemble (Fig. S12).

Since the maximum impact of the north-easterly moisture transport into SAF responsible for the wet bias in CMIP5 occurs during DJF (Munday and Washington, 2018), the impact of the CORDEX0.44 and CORDEX0.22 increase in resolution and the effect of the improved representation of topography is also more intensely identified during DJF. As it is displayed in Fig. 4, surface orography is substantially improved in the CORDEX ensembles, relative to CMIP5/6. The

improvement of orography has a further effect in blocking moisture transport entering SAF from the northeast, especially during Dec-Jan, as seen in Fig. 5.

### 3.3 Angola low

In Fig. 6 the mean monthly climatology of the AL pressure system during the rainy season is displayed for the period 1986-2005. The AL is explored by means of relative vorticity, only within the region extending from 14 °E to 25 °E and from 11 °S to 19 °S. This region is characterized by Howard and Washington (2018) as the main region of interest for the AL. The relative vorticity for $\zeta <-0.00001$ s$^{-1}$ over the whole SAF is shown in Fig. S13. In addition, potential temperature at 850 hPa (theta850) is overlaid on relative vorticity, with the first contour set at 308 K, the last contour set at 318 K and the increment between the isotherms being set to 2 K. For ERA5 and the ensemble means of CMIP5/6 the geopotential height at 850 hPa was also available.

As shown in Fig. 6, $\zeta$ values for October are greater than >-0.000025 s$^{-1}$ for ERA5 and CORDEX0.44/0.22 and are relatively weaker in CMIP5 and even weaker in CMIP6. The high cyclonic vorticity values overlap with the 312 K isotherm for all datasets. We also observe that the isoheights in the ERA5 and CMIP5/6 ensembles are closely collocated with the 312 K isotherms, indicating that the low pressure system observed over the region is caused by the excess heating of the air and hence, it is indicative of a typical heat low pressure system (Munday and Washington, 2017; Howard and Washington, 2018). Moving to November, the picture is similar however, the isotherms display a southward extension, while the 850 hPa isoheigths deepen ~5 m in ERA5 and CMIP5/6. In December, all datasets display an increase in cyclonic vorticity, however, the maximum heating area has migrated southwards over the Kalahari region. This fact indicates that cyclonic activity over the AL region is no longer due to thermal causes. During December and January the cyclonic activity is enhanced in all datasets and the isotherms have migrated even more southwards, forming the Kalahari heat low, which is distinct from the AL. We also observe that during January, the isoheights in ERA5 and CMIP5/6 become even deeper. We also note that the elongated trough during Dec-Jan can be indicative of the formation of TTTs, which account for a large proportion of rainfall over SAF (Hart et al., 2010). February displays similar spatial patterns to January for all datasets, however slightly weakened for all variables. In March, cyclonic activity over the region has seized.  Taking into consideration the distribution of the cyclonic vorticity field, we observe that in higher resolution datasets (ERA5, CORDEX0.22) high vorticity values are more severe, on very localized regions. With respect to potential temperature, we observe for October and November all datasets having a similar distribution of theta850 values. We also note that CMIP6, in general, displays higher theta850 values and lower geopotential heights, relative to CMIP5.

### 3.4 Precipitation indices

Total annual precipitation (PRCPTOT) is displayed in Fig. 7 (a). The mean of the CMIP6 ensemble displays the largest amounts of PRCPTOT (approximately 1000 mm year$^{-1}$), with CMIP5 following closely. The CORDEX0.44 and

CORDEX0.22 ensembles display a very similar behaviour, systematically reducing PRCPTOT amounts seen in CMIP5/6 by approximately 200 mm year$^{-1}$, yielding PRCPTOT values closer to that of the observational datasets. Both CMIP5/6 and CORDEX0.22/0.44 ensembles display similar within-ensemble variability. The ensemble mean of the observational datasets is considerably lower than CORDEX ensembles and displays an interannual variability between 500-800 mm year$^{-1}$. Both the ensemble means of CMIP5/6 and CORDEX0.44/0.22 fail to reproduce the interannual variability of the observational ensemble. In Fig. 7 (b) the annual maximum 1 day precipitation (Rx1Day) is displayed. For Rx1Day, the mean of the CMIP5 ensemble is in close agreement with the mean of the observational ensemble (approximately 40 mm d$^{-1}$). The ensemble mean of CORDEX0.44 yields larger precipitation amounts (approximately 55 mm d$^{-1}$) than CMIP5 and the observational ensemble. The CORDEX0.22 ensemble mean displays even higher values (approximately 75 mm d$^{-1}$). As it is shown in Fig. 7 (b), the CORDEX0.44 ensemble mean is influenced by higher Rx1Day values, originating from ensemble members that cluster within the range 65-85 mm d$^{-1}$. The spread of the CMIP5 ensemble is comparable to that of the observational data, however, the CORDEX0.44/0.22 ensemble spreads are still larger, ranging from 25-85 and from 55-100 mm d$^{-1}$, respectively. The CMIP6 ensemble falls between the CORDEX0.44 and CMIP5 ensembles, with a spread comparable to that of CMIP5. In Fig. 7 (c) the annual number of days with daily precipitation greater than 10 mm (R10mm) is presented. It is noted that the ensemble mean of the CORDEX0.44 ensemble is close to that of the observational datasets (~25 days year$^{-1}$ with daily precipitation greater than 10 mm), while the ensemble mean of CORDEX0.22 almost coincides with the mean of the observational datasets. The mean of the CMIP5 ensemble yields approximately 34 days of extreme precipitation annually. It is also highlighted that the CMIP5 ensemble displays a large range of R10mm values (10-55 days year$^{-1}$). Again, the CMIP6 ensemble mean coincides with that of CMIP5. In Fig. 7 (d) the annual number of days with daily precipitation greater than 20 mm (R20mm) is shown. There is close agreement between the CMIP5 and CORDEX0.44 ensembles, however both datasets overestimate R20mm relative to the observational data. Again, the CMIP5 ensemble displays the largest spread and a very weak interannual variability is seen on both CMIP5 and CORDEX0.44 ensemble means. The CMIP6 ensemble mean is slightly larger than its predecessor. R20mm in CORDEX0.22 mean is almost identical to the mean of the CMIP6 ensemble.

### 3.5 Trends

In Fig. 8 the monthly precipitation trends for the rainy season of the period 1986-2005 are displayed for all 3 observational data (gauge-based, SatelliteMerge Satellite-NoMerge) and for the CORDEX0.44, CORDEX0.22, CMIP5 and CMIP6 ensembles. Precipitation trends display considerable agreement among all 3 observational datasets, both concerning the signal and the magnitude of the trend. However, the CORDEX0.44/0,22 and CMIP5/6 ensembles display trends that are considerably smaller in magnitude. In addition, CORDEX0.44, CMIP5 and CMIP6 ensembles display fairly distinct spatial patterns that are not in agreement either among them, or with the spatial pattern of precipitation trends displayed by the observational datasets. In general, we observe that the signal between CORDEX0.44 and CORDEX0.22 is consistent, with trends in CORDEX0.22 displaying a larger magnitude.

More specifically, during October, all observational products display decreasing trends for the most part of SAF that reach up to -0.1 mm d$^{-1}$ per 20 years. During November the signal changes and SAF experiences increasing trends, with an exception for NW SAF, northern Mozambique and regions of eastern South Africa. During December increasing trends become even more spatially extended and pronounced, especially for satellite products. During January, certain areas of decreasing trends over northern SAF appear, while during February decreasing trends are observed over almost the whole

extent of SAF. In March, increasing trends are observed in the region extending from southern Mozambique and stretching towards Zimbabwe and southern Zambia.

Monthly precipitation trends in the CORDEX0.44 ensemble are significantly weaker than in the observational datasets and display precipitation increase during Oct–Dec. After January certain regions of intensified decreasing trends appear over southern Angola-northern Namibia and Botswana (Jan) and over Botswana and South Africa (Feb). The pattern of trends is

relatively similar in CORDEX0.22, however, the trend magnitude is more enhanced. In CMIP5 decreasing trends are observed during October, but for November increasing trends are observed over the northern part of SAF. During December, strong increasing trends (0.1 mm d$^{-1}$ per 20 years) appear for central SAF, while, during January almost all of the SAF region (with an exception for Mozambique) experiences decreasing precipitation trends. In CMIP6 persistent drying trends are observed almost throughout the whole of SAF and are particularly strong during Jan-Feb (-0.1 mm d$^{-1}$ per 20 years). During

March however, the signal is reversed. Statistical significance assessed with the Mann-Kendall test is shown in Fig. S14. The number of ensemble members displaying increasing or decreasing trends in each ensemble is shown in Fig. S15.

**4 Discussion and conclusions**

The analysis of the SD among the different observational products highlights the fact that precipitation assessment requires consultation of multiple (gauge-based, satellite and reanalysis) products. If this is not possible, then it is highly

recommended that the spatial distribution and frequency of reporting of the underlying station data is examined, for each respective precipitation product in use. This should be also regarded in cases when gauge-based or satellite products are utilized for model evaluation purposes. Moreover, satellite products that merge with rain gauges should not be considered independent from gauge-based products that exploit similar gauge-networks. In addition, we note that SD in the CORDEX0.44 ensemble is considerably lower than in the CMIP5/6 ensembles, supplying evidence that the CORDEX0.44

set of simulations provide more constrained results and can thus be considered to be a suitable dataset for climate impact assessment studies over SAF. However, that is not entirely the case for the CORDEX0.22 ensemble, which although it displays SD values smaller to that of CMIP5/6, it still yields SD values higher than that of CORDEX0.44.

Concerning the annual cycle of precipitation, we note that although the seasonality is captured reasonably by both the CMIP and CORDEX-Africa ensembles, still, there are considerable differences between them. More specifically, we

conclude that the CORDEX0.44 ensemble exhibits smaller ensemble spread for all months of the rainy season compared to the driving GCMs (CMIP5). We also conclude that the strong wet bias over SAF in the CMIP5 ensemble (Munday and

Washington, 2018) is considerably reduced in the CORDEX0.44 ensemble. This bias is still evident in CMIP6. A plethora of references in the literature (Reason and Jagadheesha, 2005; Lyon and Mason, 2007; Crétat et al., 2019; Munday and Washington, 2017; Howard and Washington, 2018) have highlighted the importance of the AL pressure system in modulating precipitation over SAF. We note that the strength of the AL as assessed in the current study was simulated to be weaker in the CORDEX0.44 than in the CORDEX0.22 ensemble. This may partly explain why precipitation in the CORDEX0.44 ensemble is reduced, relative to the CORDEX0.22 ensemble. However, there is need for a more in-depth dynamical analysis of the simulation of the AL in the CORDEX-Africa ensemble (both CORDEX0.44 and CORDEX0.22) and its impact on modulating precipitation seasonality and patterns over SAF.

The use of the 4 ETCCDI indices demonstrated that the CORDEX-Africa ensemble yields results that are in closer agreement to the observational data, compared to CMIP5/6 ensembles. It is, nevertheless, noticed that the improvement in the CORDEX-Africa ensemble is most evident when the ensemble mean is used. This highlights the fact that the ensemble mean performance is improved, relative to the performance of individual models **(Nikulin et al., 2012b)**. For this reason, it is advisable that climate impact studies employ multi-model ensemble means, as a method of obtaining the consensus climatic information emanating from various models (Duan et al., 2019). In addition, we underline the fact that in all indices the ensemble means of CMIP5/6 and CORDEX-Africa were not able to reproduce the interannual variability that was seen in the observational ensemble. This remark is in agreement with the fact that the task of reproducing precipitation variability across various time-scales by the CMIP5 ensemble is known to present challenges (Dieppois et al., 2019), that inevitably cascade into the CORDEX-Africa simulations that are forced with CMIP5 GCMs (Dosio et al., 2015). Lastly, even though the CORDEX-Africa ensembles reduce precipitation amounts over SAF, their use in drought-related impact studies should take into consideration that still, they yield larger precipitation amounts than the observational data, which might eventually lead to underestimation of drought risk.

Precipitation trends during the rainy season displayed high spatial variability depending on the month. All observed (gauge-based and satellite) trends display substantial spatial agreement. The precipitation trends obtained by the CMIP5/6, and CORDEX0.44/0.22 ensembles, did not display consistency with the trends obtained from the observational datasets. This is not entirely unexpected, due to the role of internal variability compared to external forcing in recent decades (Pierce et al., 2009), unlike temperature trends which have been shown to have a good agreement between the CORDEX-Africa (at 0.44º degrees of spatial resolution) and CMIP5 ensembles with observed temperature trends (Dosio and Panitz, 2016; Warnatzsch and Reay, 2019). Nonetheless, we note that the trend signal between CORDEX0.44 and CORDEX0.22 is consistent, with CORDEX0.22 in general enhancing the CORDEX0.44 precipitation trends.

In conclusion, while CORDEX0.44 displays marked improvement over coarser resolution products, there are still further improvements to be made. More specifically, since the wet bias in RCM simulations persists (although considerably reduced relative to GCMs), it is necessary that precipitation over southern Africa is no longer assessed based on bulk descriptive statistics, but that there will be a shift towards process-based evaluation, where the dynamical and thermodynamical characteristics of specific atmospheric features are investigated more thoroughly in the CORDEX-Africa simulations. For

this reason, it is imperative that all institutes submitting RCM simulations in data repositories such as the Earth System Grid Federation or the Copernicus Climate Data Store, provide model output data on multiple pressure levels, so that a fair comparison with the CMIP community would be possible. In addition, since the climate of southern Africa is highly coupled with the moisture transport coming from the adjacent oceans, it is necessary that the next generation of RCM simulations within CORDEX-Africa are performed coupled with ocean models. Lastly, since convection over southern Africa has a strong thermal component during specific months of the year (Oct-Nov), it is necessary that the land-atmosphere coupling processes within each RCM are examined in more detail, with coordinated efforts such as the LUCAS Flagship Pilot Study (https://ms.hereon.de/cordex_fps_lucas/index.php.en), as performed in the Euro-CORDEX domain. In the world of regional climate modelling community, the 0.44° resolution of CORDEX-Africa is no longer state of the art and ensemble efforts are now approaching convection permitting grid-spacing (i.e., < 4 km) in some parts of the world (Ban et al., 2021; Pichelli et al., 2021). We also note, that increasing effort should be made with regards to understanding the improvements made from CORDEX0.44 simulations to CORDEX0.22. Although higher resolution is a desired target in the climate modelling community due to the more realistic representation of processes that it offers, still it should not be used as a panacea. In the current work we identified certain weaknesses in the CORDEX0.22 ensemble, that should be addressed before the community populates further its simulation matrix. The next generation ensembles for Africa will hopefully provide insight and improvements to the challenges described here.

*Code and data availability*

Analysis was performed using the R Project for Statistical Computing (https://www.r-project.org/), the Climate Data Operators (CDO) (https://code.mpimet.mpg.de/projects/cdo/) and Bash programming routines. Processing scripts are available via ZENODO under DOI: https://doi.org/10.5281/zenodo.4725441. CMIP5, CMIP6 and CORDEX-Africa daily precipitation data were retrieved from the Earth System Grid Federation (ESGF) portal (https://esgf-data.dkrz.de/projects/esgf-dkrz/). CMIP5 temperature data at 850 hPa were retrieved from the Climate Data Store (CDS) (https://cds.climate.copernicus.eu/#!/home). CORDEX-Africa (both at 0.44° and 0.22° spatial resolution) temperature data at 850 hPa were retrieved from ESGF. Surface elevation data for CMIP5 and CORDEX-Africa were retrieved from ESGF. The Shuttle Radar Topography Mission (SRTM) Digital Elevation Model was retrieved from: https://srtm.csi.cgiar.org/. ERA5 data were retrieved from CDS. Climate Research Unit (CRU) data are available at: https://crudata.uea.ac.uk/cru/data/hrg/. The University of Delaware (UDEL) data are available at: https://psl.noaa.gov/data/gridded/data.UDel_AirT_Precip.html. The CPC Global Unified Gauge-Based Analysis of Daily Precipitation (CPC-Unified) was retrieved from: https://psl.noaa.gov/data/gridded/data.cpc.globalprecip.html. NOAA's PRECipitation REConstruction over Land dataset (PREC/L) was retrieved from: https://psl.noaa.gov/data/gridded/data.precl.html. The dataset of the Global Precipitation Climatology Centre (GPCC) was retrieved from: https://psl.noaa.gov/data/gridded/data.gpcc.html. The Tropical Applications of Meteorology using SATellite (TAMSAT) data were retrieved from: http://www.tamsat.org.uk/. The Precipitation

Estimation from Remotely Sensed Information using Artificial Neural Networks – Climate Data Record (PERSIANN-CDR)

are available at: https://chrsdata.eng.uci.edu/. The Climate Hazards Group InfraRed Precipitation with Station data (CHIRPS) products are available at: https://www.chc.ucsb.edu/data/chirps. The CPC Merged Analysis of Precipitation (CMAP) dataset was retrieved from: https://psl.noaa.gov/data/gridded/data.cmap.html. The Global Climatology Precipitation Project (GPCP) dataset was retrieved from: https://psl.noaa.gov/data/gridded/data.gpcp.html. The African Rainfall Climatology (ARC) dataset is available at:

https://iridl.ldeo.columbia.edu/SOURCES/.NOAA/.NCEP/.CPC/.FEWS/.Africa/.DAILY/.ARC2/.daily/index.html?Set-Language=en.

*Supplement*

The supplement related to this article is available online.

*Author contribution*

MCK, EK and SPS designed the research. MCK implemented the analysis and prepared the manuscript. EK and SPS editted the manuscript and provided corrections.

*Competing interests*

The authors declare that they have no competing interests.

*Acknowledgements*

MCK was funded by the Hellenic Foundation for Research & Innovations, under the 2nd Call for PhD Candidates
(application No. 1323). This article is funded by the AfriCultuReS project "Enhancing Food Security in African Agricultural Systems with the Support of Remote Sensing", (European Union's Horizon 2020 Research and Innovation Framework Programme under grant agreement No. 774652). The authors would like to thank the Scientific Support Centre of the Aristotle University of Thessaloniki (Greece) for providing computational/storage infrastructure and technical support.

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

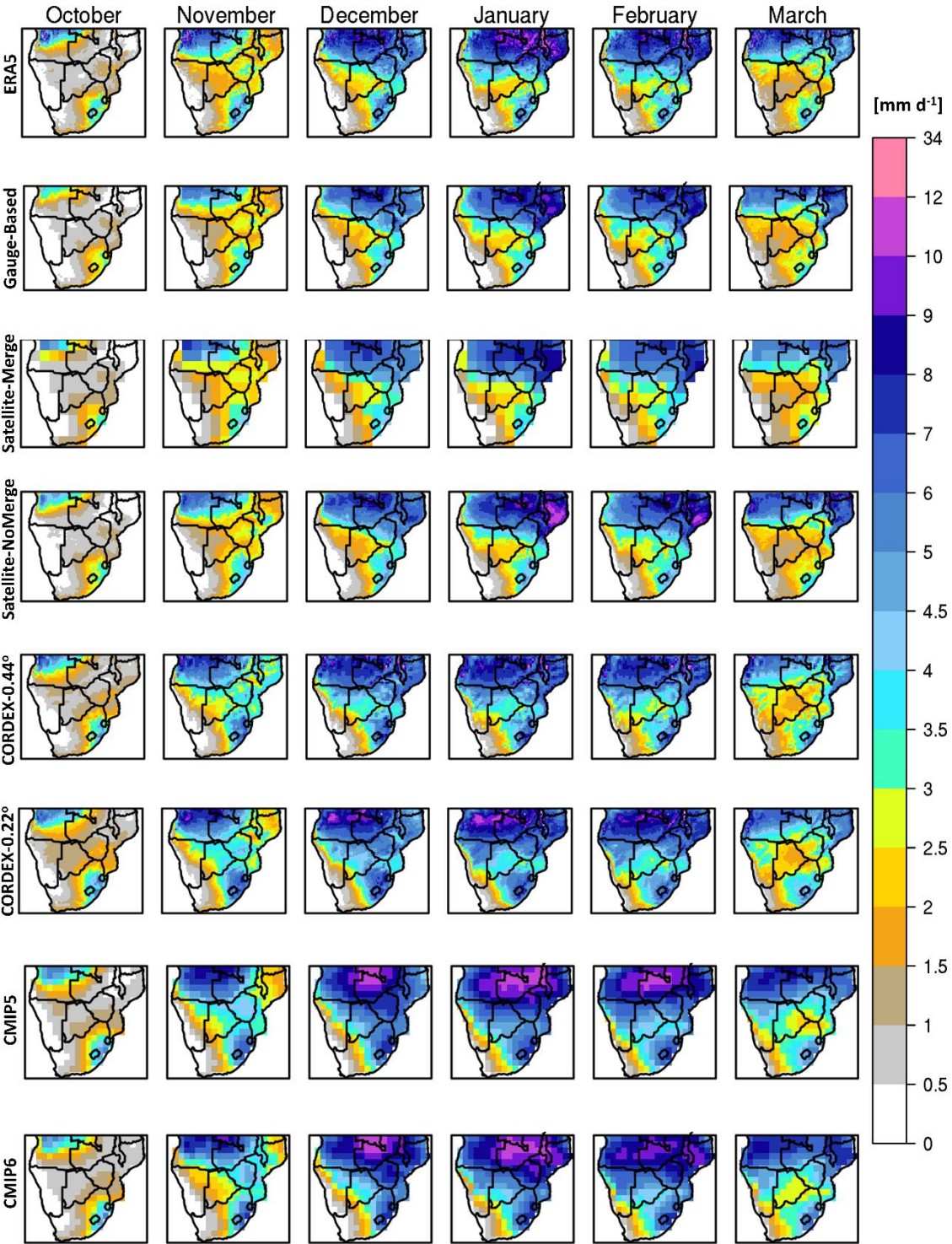

**Figure 1.** Monthly precipitation climatologies during the period 1986-2005 in mm d$^{-1}$. More specifically, from top to bottom: ERA5 reanalysis dataset. Gauge-based: Ensemble mean of datasets that were produced by employing spatial interpolation methods using rain gauges/station data. Satellite-Merge: Ensemble mean of all satellite products that merge with rain gauges/station data. Satellite-NoMerge: Ensemble mean of satellite products that do not merge with rain gauges/station data. CORDEX-0.44º: Ensemble mean of regional climate model simulations performed in the context of the Coordinated Regional Climate Downscaling Experiment (CORDEX) – Africa domain with a spatial resolution equal to 0.44º x 0.44º. CORDEX-0.22º: CORDEX-Africa simulations with a spatial resolution equal to 0.22º x 0.22º. CMIP5: Ensemble mean of general circulation models participating in the Coupled Model Intercomparison Project Phase 5 (CMIP5) that were used as forcing in the CORDEX-Africa simulations. CMIP6: Ensemble mean of general circulation models participating in the Coupled Model Intercomparison Project Phase 6.

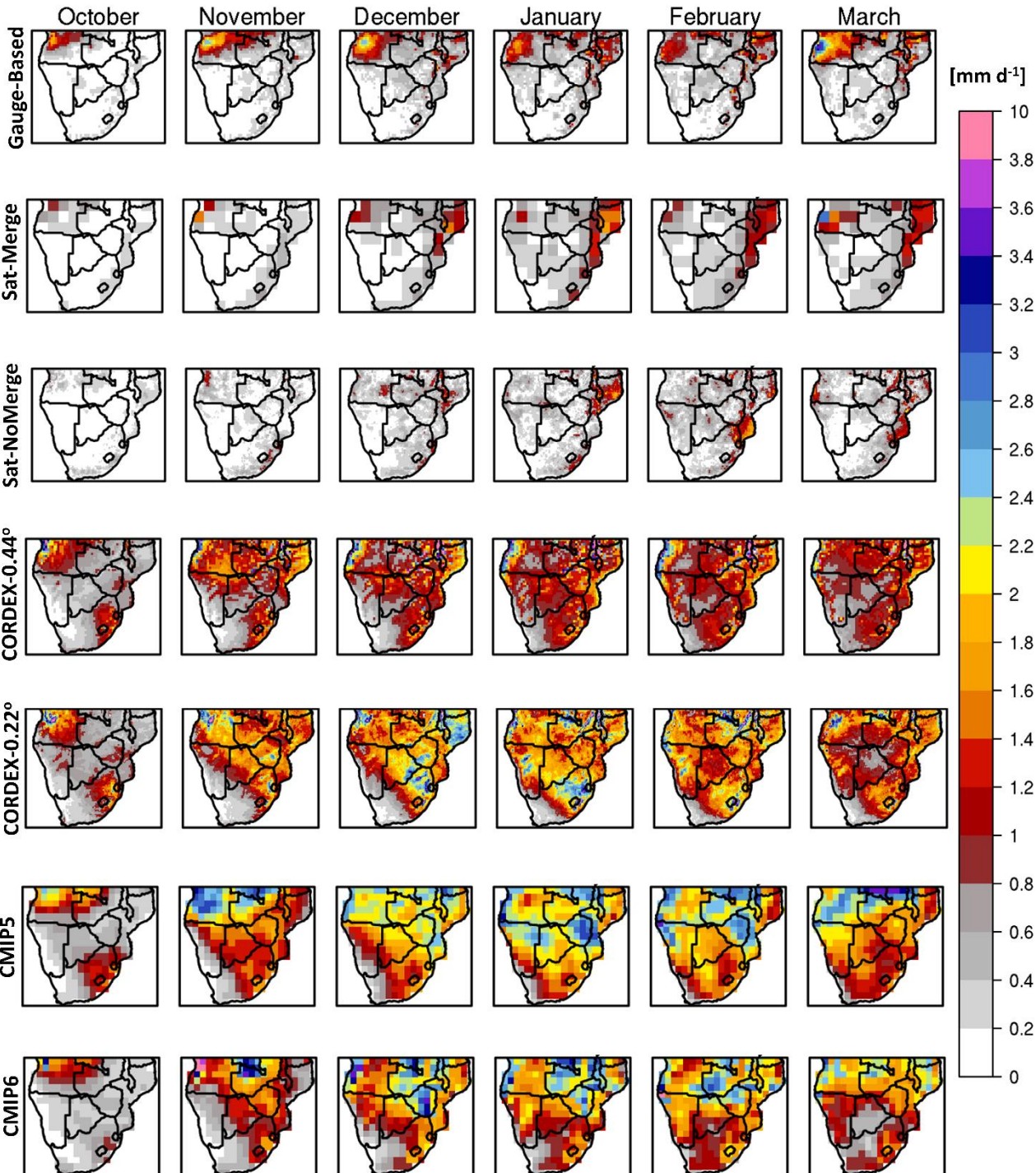

**Figure 2.** Standard deviation of monthly precipitation [mm d$^{-1}$] during the period 1986-2005. Rows indicate the ensemble means analyzed. From top to bottom: Gauge-based: Ensemble mean of datasets that were produced by employing spatial interpolation methods using rain gauges/station data. Sat-Merge: Ensemble mean of all satellite products that merge with rain gauges/station data. Sat-NoMerge: Ensemble mean of satellite products that do not merge with rain gauges/station data. CORDEX-0.44º: Ensemble mean of regional climate model simulations performed in the context of the Coordinated Regional Climate Downscaling Experiment – Africa domain with a spatial resolution equal to 0.44º x 0.44º. CORDEX-0.22º: CORDEX-Africa simulations with a spatial resolution equal to 0.22º x 0.22º. CMIP5: Ensemble mean of general circulation models participating in the Coupled Model Intercomparison Project Phase 5 (CMIP5) that were used as forcing in the CORDEX-Africa simulations. CMIP6: Ensemble mean of general circulation models participating in the Coupled Model Intercomparison Project Phase 6.








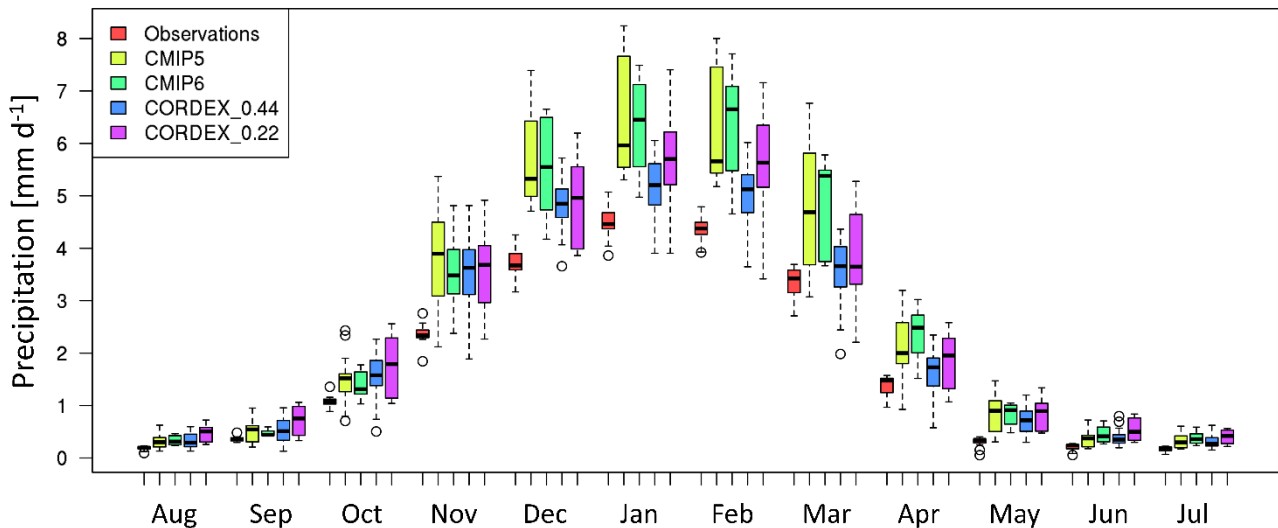

**Figure 3.** Annual cycle of monthly precipitation during 1986-2005 for the ensemble of observational data (gauge-based, satellite and reanalysis), CMIP5 (Coupled Model Intercomparison Project Phase 5), CMIP6 (Coupled Model Intercomparison Project Phase 6), CORDEX0.44 (Coordinated Regional Climate Downscaling Experiment – Africa domain with a spatial resolution equal to 0.44º x 0.44º) and CORDEX-0.22º (CORDEX-Africa simulations with a spatial resolution equal to 0.22º x 0.22º). The thick horizontal black lines indicate the ensemble median for each month, the box encloses the interquartile range, and the tails denote the full ensemble range. Circles represent the outliers for each ensemble. Grid points only are considered.

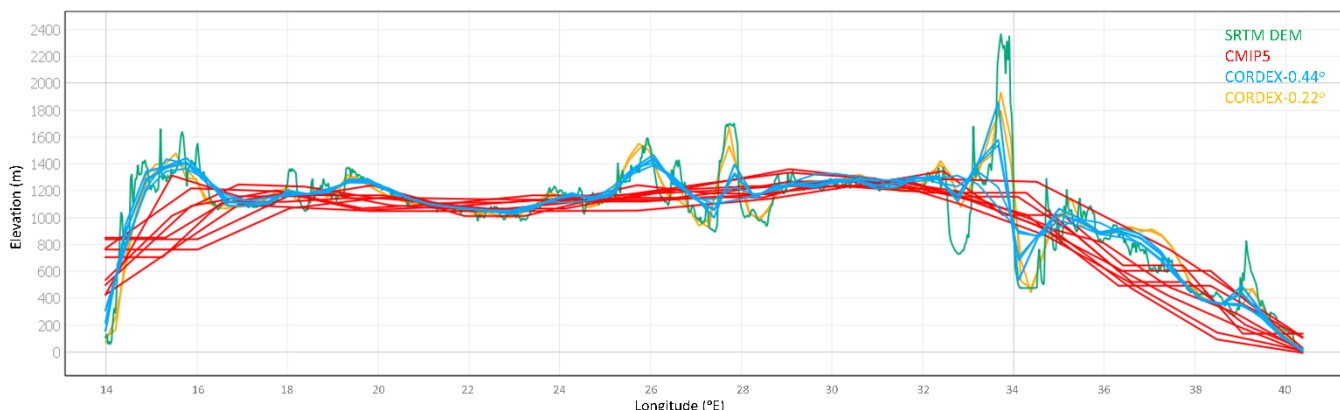

**Figure 4.** Cross section of surface elevation at 11ºS across southern Africa for the Shuttle Radar Topography Mission (SRTM) Digital Elevation Model (in green), the surface altitude as represented in the CMIP5 (Coupled Model Intercomparison Project Phase 5) global climate models (in red),the surface altitude as represented in the CORDEX0.44

(Coordinated Regional Climate Downscaling Experiment – Africa domain with a spatial resolution equal to 0.44º x 0.44º) (in blue) and the surface altitude as represented in the CORDEX-0.22º (CORDEX-Africa simulations with a spatial resolution equal to 0.22º x 0.22º) (in yellow).


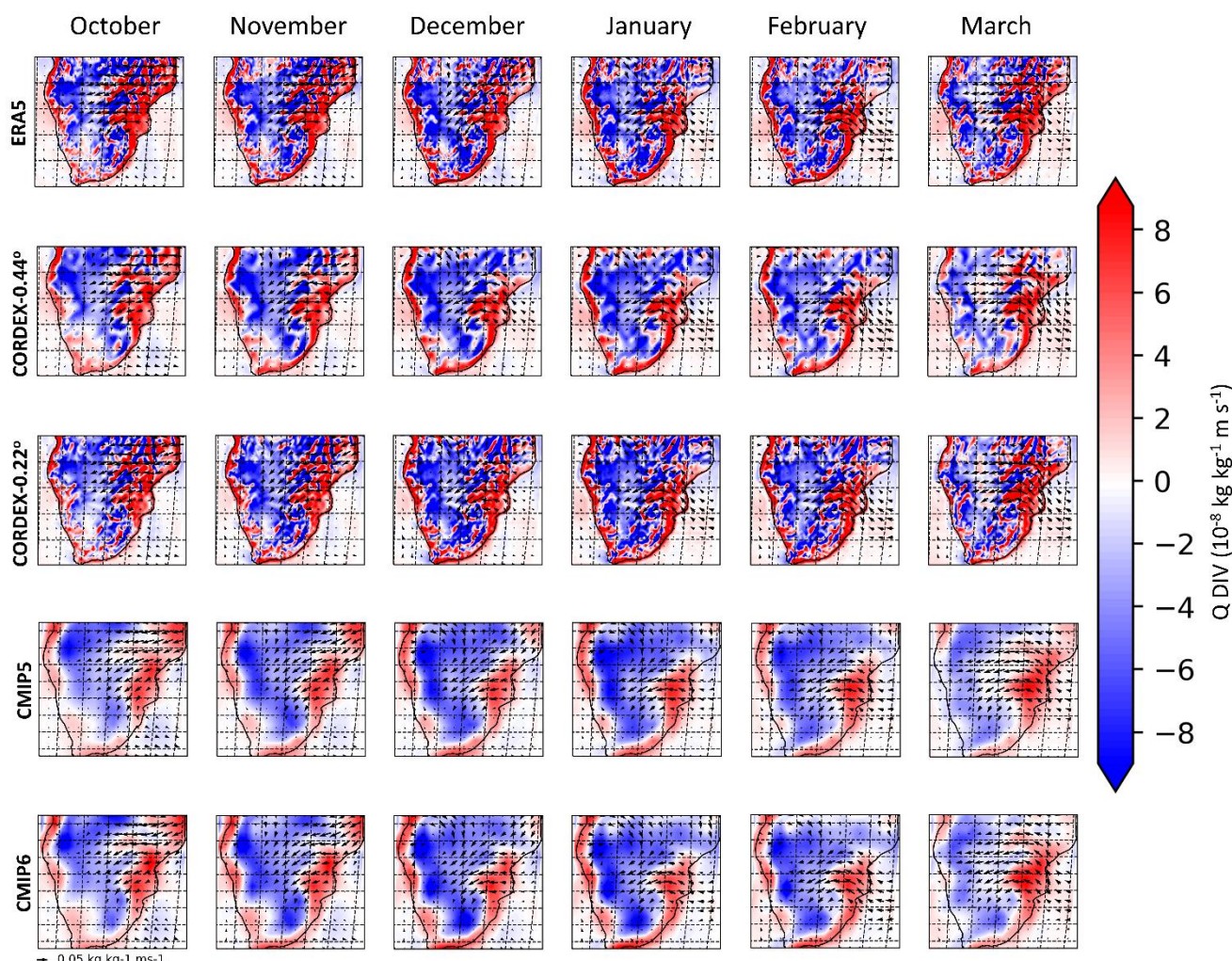

**Figure 5.** Mean monthly moisture flux and divergence at 850 hPa during the period 1986-2005. Rows indicate the ensemble means analyzed. From top to bottom: ERA5, ensemble mean of CORDEX0.44º, CORDEX0.22º, CMIP5 and CMIP6 simulations.


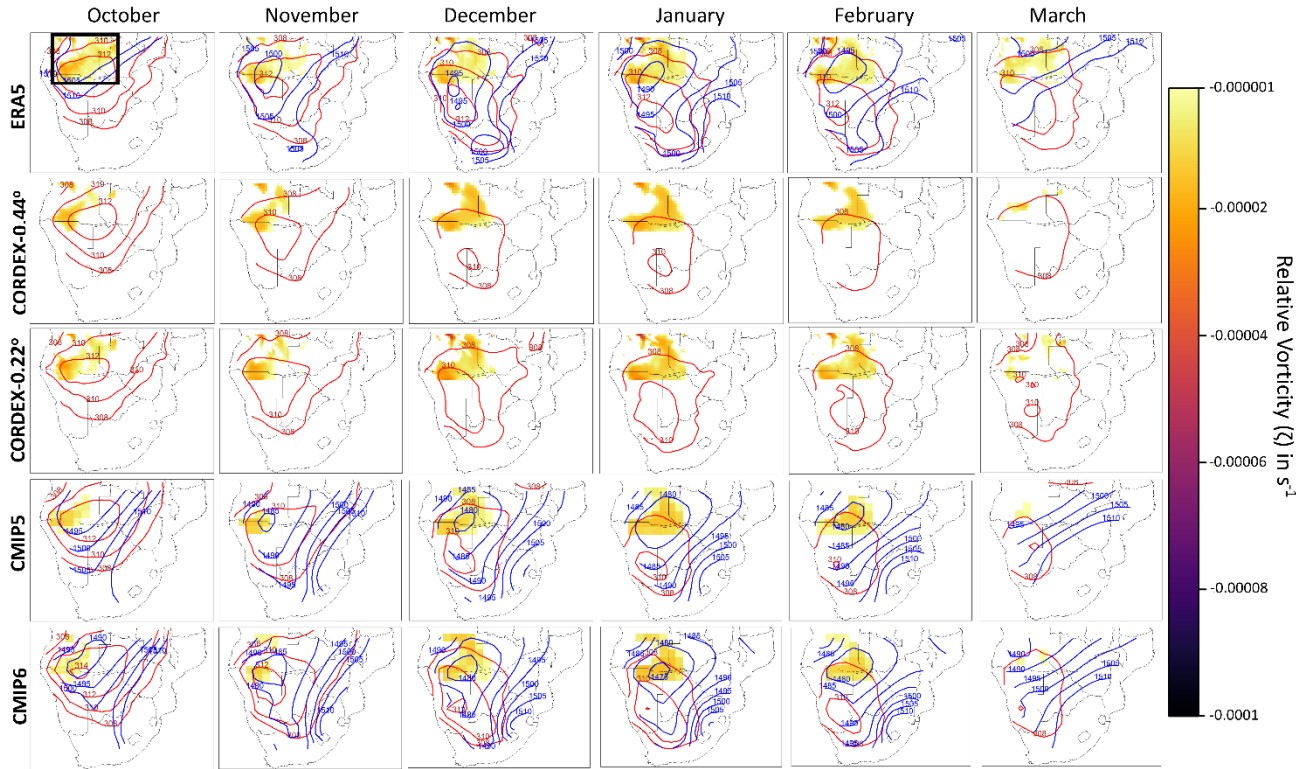

**Figure 6.** Monthly climatologies of the Angola Low pressure system during the rainy season for the period 1986-2005. Filled contours indicate cyclonic relative vorticity ($\zeta$) for $\zeta < -0.00001$ s$^{-1}$ over the region extending from 14 ºE to 25 ºE and from 11 ºS to 19 ºS. Red lines indicate the isotherms of potential temperature at 850 hPa, having an increment of 2 K. Blue lines indicate isoheights of the geopotential height at 850 hPa, having an increment of 5 m. CORDEX0.44/0.22 are not
plotted with geopotential isoheights, because this variable was not available for CORDEX simulations. From top to bottom: ERA5, ensemble mean of CORDEX0.44º, CORDEX0.22º, CMIP5 and CMIP6 simulations. Black box indicates the region from 14 ºE to 25 ºE and from 11 ºS to 19 ºS.

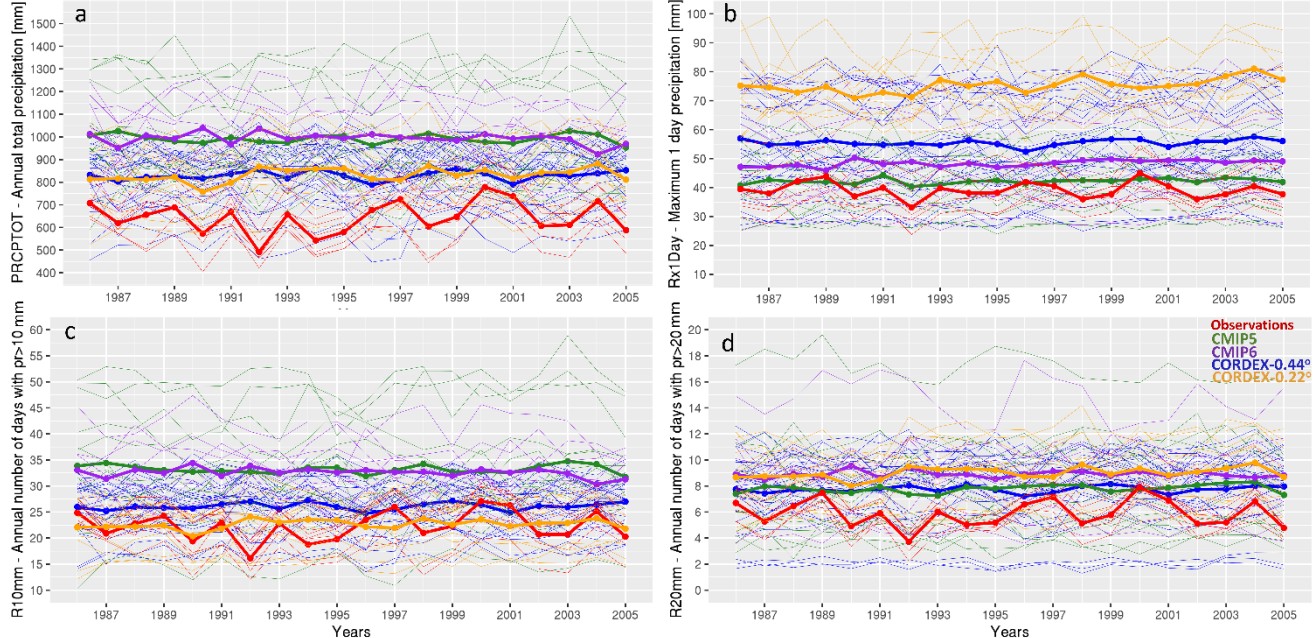

**Figure 7.** Timeseries of the ETCCDI indices over southern Africa (10 ºE to 42 ºE and 10 ºS to 35 ºS) for the observational ensemble in red (gauge-based, satellite and reanalysis), CMIP5 (Coupled Model Intercomparison Project Phase 5) ensemble in green, CMIP6 (Coupled Model Intercomparison Project Phase 6) ensemble in purple, CORDEX-0.44º: Ensemble mean of regional climate model simulations performed in the context of the Coordinated Regional Climate Downscaling Experiment – Africa domain with a spatial resolution equal to 0.44º x 0.44º in blue and CORDEX-0.22º in orange. Thin lines display single ensemble members, thick lines display ensemble means. Y-axis on each panel depicts: (a) PRCPTOT (total annual precipitation), (b) Rx1Day (annual maximum daily precipitation), (c) R10mm (annual number of days with daily precipitation >10 mm), (d) R20mm (annual number of days with daily precipitation >20 mm).

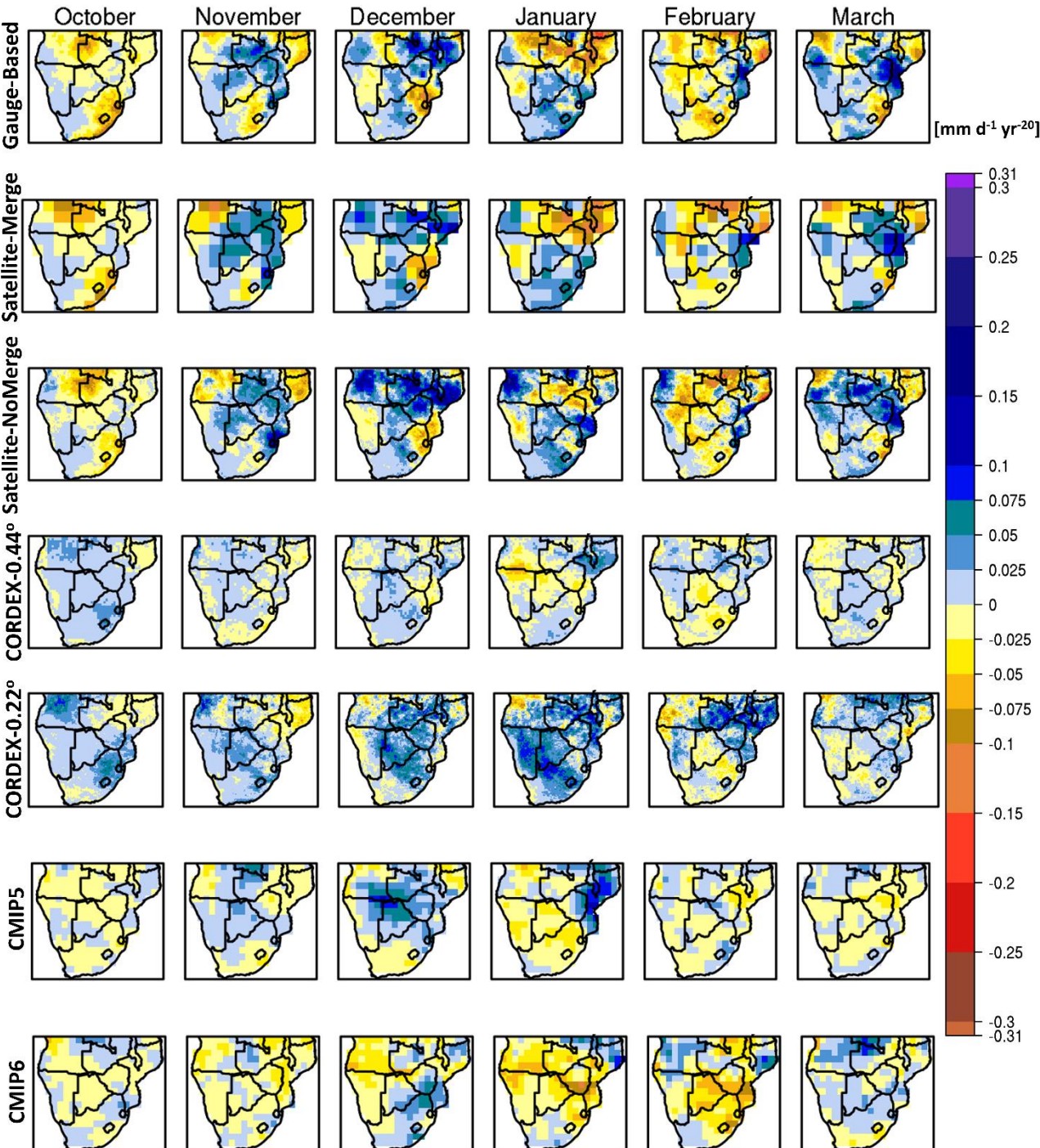

**Figure 8.** Trends for monthly precipitation for the period 1986-2005 [mm d$^{-1}$ per 20 years] calculated using Sen's Slope. Rows indicate the ensemble mean of trends produced by each ensemble member. From top to bottom: Gauge-Based: Ensemble mean of datasets that were produced by employing spatial interpolation methods using rain gauges/station data. Satellite-Merge: Ensemble mean of all satellite products that merge with rain gauges/station data. Satellite-NoMerge: Ensemble mean of satellite products that do not merge with rain gauges/station data. CORDEX-0.44$^o$: Ensemble mean of regional climate model simulations performed in the context of the Coordinated Regional Climate Downscaling Experiment – Africa domain with a spatial resolution equal to 0.44$^o$ x 0.44$^o$. CORDEX-0.22$^o$: CORDEX-Africa simulations with a spatial resolution equal to 0.22$^o$ x 0.22$^o$. CMIP5: Ensemble mean of general circulation models participating in the Coupled Model Intercomparison Project Phase 5 (CMIP5) that were used as forcing in the CORDEX-0.44$^o$ simulations. CMIP6: Ensemble mean of general circulation models participating in the Coupled Model Intercomparison Project Phase 6.