# Peer review of "Precipitation over southern Africa: Is there consensus among GCMs, RCMs and observational data?"

_Geoscientific Model Development, 2021_

## Author Comment (AC1)

**Responses to Anonymous Reviewer 1**

**General Comment:**

The paper presents an analysis of precipitation in Southern Africa from RCMs, CMIP5 and 6, GCMs and observational data during a historical period from 1986-2005. The authors make use of several groupings of RCMs, GCMs and observational data into ensembles for a more thorough analysis. The focus of the paper is on the differences in annual and monthly rainfall in the southern Africa region between the different sets of ensembles and in terms of various metrics of rainfall. The paper also compares the trend in rainfall over this historical period between the RCMs, GCMs, and observational ensembles to understand the fidelity of the models compared to observed datasets in this region. In general, I find the paper well sourced and written however there are some instances where I find the wording confusing. I find the scientific analysis of the paper quite sound and thorough. My main concerns with the paper are that the novelty of the paper within the context of model development is not clearly defined. I also find the analysis and conclusions quite general and I think the focus of the paper could be improved.

**RESPONSE**: We thank very much Anonymous Reviewer #1 for this comment! We have followed closely their corrections and recommendations and we have addressed all points stated.

**Major comments:**

**1ˢᵗ Comment:**

I recommend more clearly laying out what the novelty or newness is of this work. Based on previous work it seems that precipitation in this region has been studied in similar ways before. Why is the method/results/approach in this work an improvement on those studies?

**RESPONSE**: Thank you very much for this comment. Indeed, precipitation over southern Africa has been studied before. More specifically, Nikulin et al. (2012) was the first to present an overview of the CORDEX-Africa ensemble and to analyze the spatiotemporal patterns of precipitation. They showed that during the rainy season (Jan-Mar as used in Nikulin et al. (2012)) there is a weak wet bias over southern Africa, and that the use of the ensemble mean was able to outperform individual models, highlighting the importance of ensemble-based approaches. The Nikulin et al. (2012) analysis was conducted on a pan-African scale. Similarly, Kalognomou et al., (2013) analyzed the same ensemble of CORDEX-Africa simulations, focusing over southern Africa and reported similar findings to Nikulin et al., (2012). In Shongwe et al. (2014) a particular emphasis was put on the onset and retreat of the rainy season, especially over the eastern part of southern Africa. Nonetheless, as stated in Shongwe et al. (2014) "No attempt is made in this paper to identify the model physics and dynamics responsible for the differences in RCM performance." All the aforementioned studies employed the evaluation (hindcast) simulations performed within CORDEX-Africa, driven by ERA-Interim; the analyzed ensemble was comprised of 10 RCMs. It is also worth mentioning that the regional climate model (RCM) versions used in the studies listed above, refer to previous versions of the respective RCMs, which have now been replaced by newer versions in more recent studies.

In Meque and Abiodun (2015) the same ensemble of 10 hindcast simulations was again used, but it was also compared with a set of CMIP5 GCM simulations, with the purpose to identify a causal association between ENSO and drought events over southern Africa. In Meque and Abiodun (2015) it was stated for the first time that RCMs were able to provide added value, compared to their driving GCMs. The issue of the added value of the CORDEX-Africa ensemble was clearly stated in Dosio et al. (2015), where 1 RCM participating in CORDEX-Africa (CCLM) was compared against 4 different driving GCMs. In Favre et al. (2016) a special focus was given on the annual cycle of precipitation over South Africa, using the same ensemble of 10 CORDEX-Africa hindcast simulations and in Abba Omar and Abiodun (2017), although the same hindcast ensemble was used, there was an effort to associate extreme precipitation events with dynamical processes such as the Tropical Temperate Troughs.

A comprehensive assessment of the added value between historical CORDEX-Africa RCMs simulations and of their driving CMIP5 GCMs on a seasonal timescale over the whole of Africa, was performed in Dosio et al. (2019). The first time the CORDEX-Africa

ensemble over southern Africa was compared with a plethora of observational and satellite products was presented in Abiodun et al., (2020), while the first time that CORDEX Africa at 0.44º and at 0.22º was analyzed compared to both CMIP5 and CMIP6 ensembles is presented in Dosio et al. (2021). More specifically, in Dosio et al. (2021) the analysis is performed on a seasonal timestep and on pan-African scale and its particular emphasis is placed on the projected changes of future precipitation, although a part of the analysis is dedicated to the period 1981-2010.

Our work aims to provide a comprehensive overview of the observed precipitation climatology particularly focusing over southern Africa, in all tools that are currently available in the climate community. For this reason, we employ all four ensembles used in Dosio et al. (2021) for the period 1986-2005 and we additionally employ a set of 12 observational (satellite, gridded and reanalysis) products. By doing so, we aim to highlight the precipitation uncertainty that exists even among different observational products, which is inherent in the methods used for their production. In addition, we attempt to make a connection between monthly precipitation climatology over southern Africa and a particularly important atmospheric feature, the Angola Low pressure system. To our knowledge, the Angola Low pressure system has not been studied yet within the context of CORDEX-Africa simulations. Although there has been an ample work of evaluating CORDEX-Africa simulations, we think that in order to better understand the reasons why RCM simulations do, or do not, display an improvement relative to their driving GCMs, there must be a shift towards process-based evaluations that examine particular (thermo)dynamic atmospheric processes over specific regions and specific time periods. For this reason, we also chose to perform our analysis on a monthly timescale during the rainy season (Oct-Mar). Often, seasonal means are conveniently used, however, seasonal averages might obscure spatio-temporal patterns that can only be identified on a finer temporal resolution. One of the main hindrances that often limits the ability to perform dynamic analysis in CORDEX-Africa (and CORDEX in general) simulations is the lack of available variables at different pressure levels. This was a shortcoming in our analysis also. In addition, we discuss the results with respect to monthly precipitation trends, as seen in all observational and modeling ensembles we use.

We agree that we need to present more clearly in the manuscript what the novelty and the newness of this work is. For this reason, we made the following changes in the Introduction:

Section to last paragraph: "Therefore, in this paper we expand on previous research to investigate how monthly precipitation during the rainy season over southern Africa is simulated by different modelling systems, by analyzing the monthly precipitation climatologies, the interannual variability, specific precipitation indices and monthly precipitation trends during the period 1986-2005, in four different modeling systems (CORDEX 0.22º/0.44º, CMIP5/6) and observational ensembles (satellite, reanalysis and gridded datasets). Our main goal is to provide a comprehensive overview with regards to precipitation climatology over SAF as simulated by the state-of-the-art tools used by

climate scientists. In addition, we investigate whether higher resolution models are able to provide an improved representation of precipitation over southern Africa and we investigated how a particularly important atmospheric feature, the Angola Low pressure system, is simulated in the RCM and GCM ensembles."

References:

Abba Omar, S., Abiodun, B.J., 2017. How well do CORDEX models simulate extreme rainfall events over the East Coast of South Africa? Theor. Appl. Climatol. 128, 453–464. https://doi.org/10.1007/s00704-015-1714-5

Abiodun, B.J., Mogebisa, T.O., Petja, B., Abatan, A.A., Roland, T.R., 2020. Potential impacts of specific global warming levels on extreme rainfall events over southern Africa in CORDEX and NEX-GDDP ensembles. Int. J. Climatol. 40, 3118–3141. https://doi.org/10.1002/joc.6386

Dosio, A., Jones, R.G., Jack, C., Lennard, C., Nikulin, G., Hewitson, B., 2019. What can we know about future precipitation in Africa? Robustness, significance and added value of projections from a large ensemble of regional climate models. Clim. Dyn. 53, 5833–5858. https://doi.org/10.1007/s00382-019-04900-3

Dosio, A., Jury, M.W., Almazroui, M., Ashfaq, M., Diallo, I., Engelbrecht, F.A., Klutse, N.A.B., Lennard, C., Pinto, I., Sylla, M.B., Tamoffo, A.T., 2021. Projected future daily characteristics of African precipitation based on global (CMIP5, CMIP6) and regional (CORDEX, CORDEX-CORE) climate models. Clim. Dyn. 57, 3135–3158. https://doi.org/10.1007/s00382-021-05859-w

Dosio, A., Panitz, H.-J., Schubert-Frisius, M., Lüthi, D., 2015. Dynamical downscaling of CMIP5 global circulation models over CORDEX-Africa with COSMO-CLM: evaluation over the present climate and analysis of the added value. Clim. Dyn. 44, 2637–2661. https://doi.org/10.1007/s00382-014-2262-x

Favre, A., Philippon, N., Pohl, B., Kalognomou, E.-A., Lennard, C., Hewitson, B., Nikulin, G., Dosio, A., Panitz, H.-J., Cerezo-Mota, R., 2016. Spatial distribution of precipitation annual cycles over South Africa in 10 CORDEX regional climate model present-day simulations. Clim. Dyn. 46, 1799–1818. https://doi.org/10.1007/s00382-015-2677-z

Kalognomou, E.-A., Lennard, C., Shongwe, M., Pinto, I., Favre, A., Kent, M., Hewitson, B., Dosio, A., Nikulin, G., Panitz, H.-J., Büchner, M., 2013. A Diagnostic Evaluation of Precipitation in CORDEX Models over Southern Africa. J. Clim. 26, 9477–9506. https://doi.org/10.1175/JCLI-D-12-00703.1

Kim, Y.-H., Min, S.-K., Zhang, X., Sillmann, J., Sandstad, M., 2020. Evaluation of the CMIP6 multi-model ensemble for climate extreme indices. Weather Clim. Extrem. 29, 100269. https://doi.org/10.1016/j.wace.2020.100269

Meque, A., Abiodun, B.J., 2015. Simulating the link between ENSO and summer drought in Southern Africa using regional climate models. Clim. Dyn. 44, 1881–1900. https://doi.org/10.1007/s00382-014-2143-3

Nikulin, G., Jones, C., Giorgi, F., Asrar, G., Büchner, M., Cerezo-Mota, R., Christensen, O.B., Déqué, M., Fernandez, J., Hänsler, A., Meijgaard, E. van, Samuelsson, P., Sylla, M.B., Sushama, L., 2012. Precipitation Climatology in an Ensemble of CORDEX-Africa Regional Climate Simulations. J. Clim. 25, 6057–6078. https://doi.org/10.1175/JCLI-D-11-00375.1

Shongwe, M.E., Lennard, C., Liebmann, B., Kalognomou, E.-A., Ntsangwane, L., Pinto, I., 2014. An evaluation of CORDEX regional climate models in simulating precipitation over Southern Africa. Atmospheric Sci. Lett. 16, 199–207. https://doi.org/10.1002/asl2.538

Wyser, K., van Noije, T., Yang, S., von Hardenberg, J., O'Donnell, D., Döscher, R., 2020. On the increased climate sensitivity in the EC-Earth model from CMIP5 to CMIP6. Geosci. Model Dev. 13, 3465–3474. https://doi.org/10.5194/gmd-13-3465-2020

**2ⁿᵈ Comment:**

In the second to last paragraph of the introduction the purpose and goals of the paper are given but there are several different statements of goals which I find somewhat unfocused. Is there a main goal that can be defined? It seems that the main focus of the paper is on how the RCM ensemble can be shown to be more useful for precipitation projections over this region compared to the GCMs but this is not clear. From the abstract it is also not very clear what are the main results the reader should see.

**RESPONSE**: Thank you very much for this comment and correction. Our analysis has two main goals: The first goal is to provide an intercomparison of how monthly precipitation during the rainy season over southern Africa is simulated by different modelling systems (CORDEX 0.22°/0. 44°, CMIP5/6) and to also provide an overview of the spread that is seen even among the so called "observational products", highlighting the need for improved modeling and monitoring efforts over the region. Our second goal is indeed, to highlight how RCMs are able to address certain deficiencies identified in GCMs. The second to last paragraph of the introduction has now changed to the following:

"Therefore, in this paper we expand on previous research to investigate how monthly precipitation during the rainy season over southern Africa is simulated by different modelling systems, by analyzing the monthly precipitation climatologies, the interannual variability, specific precipitation indices and monthly precipitation trends during the period 1986-2005, in four different modeling systems (CORDEX 0.22°/0.44°, CMIP5/6) and observational ensembles (satellite, reanalysis and gridded datasets). Our main goal is to provide a comprehensive overview with regards to precipitation climatology over SAF as simulated by the state-of-the-art tools used by climate scientists. In addition, we investigate whether higher resolution models are able to provide an improved representation of precipitation over southern Africa and we investigated how a particularly important atmospheric feature, the Angola Low pressure system, is simulated in the RCM and GCM ensembles."

**3rd Comment:**

My understanding is that the CORDEX-Africa 0.22° data are available. If so, why was the older 50km dataset used when a newer one was available?

**RESPONSE**: Thank you for this comment. We have now included all the CORDEX-Africa 0.22° simulations available in the analysis. The CORDEX-Africa 0.22° simulations added, are listed in the table below. We have kept, however, all CORDEX-Africa 0.44° simulations, since they constitute a larger ensemble (26 ensemble members used).

| Driving GCMs | RCMs | Realisations | Variables |
|---|---|---|---|
| CanESM2 | CanRCM4 | r1i1p1 | Pr, |
| HadGEM2-ES | CCLM5-0-15 | r1i1p1 | Pr, hus850, ua850, va850, ta850 |
| | REMO2015 | r1i1p1 | Pr, hus850, ua850, va850, ta850 |
| | RegCM4-7 | r1i1p1 | Pr, hus850, ua850, va850, ta850 |
| MPI-ESM-LR | CCLM5-0-15 | r1i1p1 | Pr, hus850, ua850, va850, ta850 |
| | REMO2015 | r1i1p1 | Pr, hus850, ua850, va850, ta850 |
| | RegCM4-7 | r1i1p1 | Pr, hus850, ua850, va850, ta850 |
| NorESM1-M | CCLM5-0-15 | r1i1p1 | Pr, hus850, ua850, va850, ta850 |
| | REMO2015 | r1i1p1 | Pr, hus850, ua850, va850, ta850 |
| | RegCM4-7 | r1i1p1 | Pr, hus850, ua850, va850, ta850 |

**Minor Comment:**

 **1st Comment:**

Line 17: SAF hasn't been defined yet, it should be defined here.

RESPONSE: We have defined SAF in line 10 (beginning of the abstract).

**2nd Comment:**

Lines 22-23: "…a similar behavior to CMIP5, however reducing slightly the ensemble spread." I would replace 'however' here with 'but'.

RESPONSE: Thank you! We will make this change in the manuscript.

**3rd Comment:**

Line 61: Over what period is this trend seen? I assume it's a historical period but it would be good to explicitly say it here.

RESPONSE: Thank you! We will make this clarification in the manuscript. The trend is calculated over the period 1986-2005. The sentence in the manuscript will be changed to "During DJF, precipitation trends for the period 1986-2005 over SAF display…".

**4th Comment:**

Sentence starting at Line 71 "However,…": This sentence is a little bit confusing I would recommend removing 'still' and the comma between 'period' and 'persist'.

RESPONSE: Thank you! The sentence in the manuscript will be changed to "However, although the CMIP6 ensemble exhibits multiple improvements on various levels (Wyser et al., 2020), certain biases and challenges identified in CMIP5 during the historical period persist in CMIP6 (Kim et al., 2020)."

**5th Comment:**

Line 90: Provide more detail of what will be addressed in the results section (Section 3). For instance, describe the subsections of the results and what will be covered.

**RESPONSE**: Thank you very much for this comment! This sentence in the manuscript will be changed to: "In Section 3 the results are presented. More specifically, the results are analyzed based on the monthly climatology, the annual cycle of precipitation, the Angola Low pressure system, the ETCCDI precipitation indices and the monthly precipitation trends. Lastly, in Section 4 we provide the discussion of the analysis along with some concluding remarks."

**6th Comment:**

Line 107: Should this be "less **than** or equal to"?

**RESPONSE**: Thank you very much. We will change the sentence to "The gauge-based products were chosen so that they have a spatial resolution less than or equal to 0.5º x 0.5º…"

**7th Comment:**

Line 183: How was the calculation of standard deviation done to get the within-ensemble agreement? Was the monthly mean of over the 1986-2005 period calculated for each model first and then the SD of the ensemble taken?

**RESPONSE**: Yes, we first calculated the monthly means over the period 1986-2005 for each model (or observational dataset) separately, and then we calculate the standard deviation.

**8th Comment:**

Figures 1, 2 and 7: The alignment and spacing of the panels is not consistent. I recommend making sure the Figures have consistent spacing and are aligned to improve their visual aesthetic.

**RESPONSE**: Thank you! We will make this correction in all panel plots!

**9th Comment:**

Lines 355-356: Expand on what improvements can be made. This is an important statement for readers who may be interested in expanding on this work.

**RESPONSE:** Thank you very much for this comment. The paragraph in the manuscript has been changed to the following: "In conclusion, while CORDEX-Africa displays marked improvement over coarser resolution products, there are still further improvements to be made. More specifically, since the wet bias in RCM simulations persists (although considerably reduced relative to GCMs), it is necessary that precipitation over southern Africa is no longer assessed based on bulk descriptive statistics, but that there will be a shift towards process-based evaluation, where the dynamical and thermodynamical characteristics of specific atmospheric features are investigated more thoroughly in the CORDEX-Africa simulations. For this reason, it is imperative that all institutes submitting RCM simulations in data repositories such as the Earth System Grid Federation or the Copernicus Climate Data Store, provide model output data on multiple pressure levels, so that a fair comparison with the CMIP community would be possible. In addition, since the climate of southern Africa is highly coupled with the moisture transport coming from the adjacent oceans, it is necessary that the next generation of RCM simulations within CORDEX-Africa are performed coupled with ocean models. Lastly, since convection over southern Africa has a strong thermal component during specific months of the year, it is necessary that the land-atmosphere coupling processes within each RCM are examined in more detail, with coordinated efforts such as the LUCAS Flagship Pilot Study (https://ms.hereon.de/cordex_fps_lucas/index.php.en), as performed in the Euro-CORDEX domain. In the world of regional climate modelling community, the 0.44º resolution of CORDEX-Africa is no longer state of the art and ensemble efforts are now approaching convection permitting grid-spacing (i.e., < 4 km) in some parts of the world (Ban et al., 2021; Pichelli et al., 2021). The next generation ensembles for Africa will hopefully provide insight and improvements to the challenges described here."

---

## Author Comment (AC2)

**Responses to Anonymous Reviewer 2**

**General Comment:**

This paper evaluates the representation of the southern African rainfall in the GCMs and RCMs compared to a set of observational data. The rainfall climatology, annual cycle, trends and a couple of ETCCDI indices are analyzed along with the representation of the Angola Low, which is one of the important driving circulations that affect the rainfall in the area. The paper is of high importance for model improvement. However, I suggest the following comments to be addressed before the paper is published in GMD.

**RESPONSE**: We would like to thank the Anonymous Reviewer #2 for the positive interpretation of the manuscript. Based on the suggestions and comments, we provide the following replies.

**Major comments:**

**1st Comment:**

**1st Comment:**

Page 8, 235-240, an evaluation of the moisture transported through the north-easterly monsoon should be performed here to support the hypothesis that the improved representation of the topography led to a lower bias in the CORDEX models.

**RESPONSE**: Thank you very much for this comment. We now include the following figure in the main manuscript, displaying the moisture flux and moisture flux divergence at 850 hPa during each month of the rainy season, for the period 1986-2005. More specifically, the moisture flux divergence was calculated using the product of specific humidity and wind at 850 hPa, following the equation below (the vertical component ($\frac{\partial qw}{\partial z}$) is considered negligible).

$$\nabla \cdot q\vec{u} = \frac{\partial qu}{\partial x} + \frac{\partial qv}{\partial y}$$

With this plot we aim to contribute to the discussion developed in Figure 11 in Munday and Washington, (2017). More specifically, one of the reasons responsible for the wet bias of CMIP5 models over southern Africa (SAF), was that mountainous regions over the northeast part of SAF were underrepresented, due to the spatial resolution of the CMIP5 models Munday and Washington, (2018). The high elevation areas over Malawi and Tanzania were not represented accurately in CMIP5 GCMs, which allowed moisture transport entering SAF from the northeast to penetrate central SAF, rather than to recurve around the high mountains and result to large precipitation amounts over northern Madagascar. Since the underrepresentation of topography in GCMs is a matter of spatial resolution, we make the hypothesis that in high resolution RCMs this issue is resolved, since moisture entering SAF from the northeast is blocked by the adequately high elevation over the Tanzania and Malawi region.

As seen in the Figure 1 below, during all months the moisture flux field is very spatially inhomogeneous in ERA5 and in both CORDEX ensembles, while in CMIP5/6 the field is considerably smoother, indicating that in low resolution GCMs the surface characteristics are not detailed enough, so as to allow for adequate friction and cause the moisture fluxes to recurve around mountainous areas. Particularly during December and January when the north-easterly monsoon is intensified, the moisture flux at the northeast of SAF is intercepted in both CORDEX ensembles, however not in CMIP5/6. After February the atmospheric flow from the northeast is weakened and it is strengthened at the southeastern part, entering SAF through Mozambique. This moisture transport originates from the Mascarene High that has developed over the South Indian Ocean. The recurvature of moisture seen at the south-eastern part of Mozambique is caused by the Mozambique Channel Trough (Barimala et al., 2018).

In the manuscript we comment concerning the moisture transport entering SAF from the northeastern part, by adding the following text as the last sentence of paragraph 3 in Section 3.2: "The improvement of orography has a further effect in blocking moisture transport entering SAF from the northeast, especially during Dec-Jan, as seen in Fig. 5."

[Figure]

Figure 1: Moisture flux and divergence at 850 hPa.

**2nd Comment:**

Page 9, section 3.3. It should be made clear why there is a special focus on the Angola low given the different processes that significantly affect the rainfall in the area. For example, the cloudband or tropical temperate trough is one of the major processes that drive rainfall in SAF but is never mentioned here. I would even suggest including the cloudbands in the analyzes.

**RESPONSE**: Thank you very much for this comment. Indeed, not mentioning the Tropical Temperate Troughs (TTTs) in the manuscript is a significant lack, since TTTs are one of the main mechanisms producing precipitation over southern Africa. We now refer to the role they play for precipitation over southern Africa in the introduction, and also in the results section (section 3.3), where findings about the Angola Low have further implications for the formation of TTTs.

More specifically, the reason why we chose to put an emphasis on the Angola Low pressure system is that usually Angola Low events precede the formation of TTTs and hence, they can be considered as their precursor in the "climate process chain" controlling precipitation over southern Africa (Daron et al., 2019). As stated in Howard et al., 2018, it is common that Angola Low events precede TTT events, since the Angola Low pressure system functions as a key process necessary for the transport of water vapor from the tropics towards the extratropics (Hart et al., 2010).

In addition, based on a Scopus query investigating the number of documents with the keywords "Angola Low" and "Tropical Temperate Troughs" in the Title-Abstract-Keywords, we saw that TTTs have received almost the double attention in the literature (47 published papers), relative to the Angola Low (23 published papers). Hence, our work is, in part, an attempt to address this gap, considering the limitations set by the availability of variables in all the ensembles that are currently examined (CORDEX-Africa 0.22º/0.44º and CMIP5/6). For this reason, we did not include an analysis of the TTTs, since it is beyond of the scope of the current study, but it is imperative that a comparative analysis of how TTTs are simulated in CORDEX-Africa 0.22º/0.44º and CMIP5/6 is performed.

**3ʳᵈ Comment:**

Page 9, section 3.3. I wonder why theta850 is used to calculate the Angola low instead of the geopotential height (as in Munday et al., 2017) or the vorticity (as in Howard et al., 2018). The CMIP6 models do have these variables available and should be used for a fair comparison.

**RESPONSE**: Indeed, Munday and Washington, (2017) use the lowest 5% of mean DJF geopotential height at 850 hPa (zg850) over southern Africa. The reason why we were not able to use the same index in order to identify the Angola Low, was that within the context of CORDEX-Africa simulations, geopotential height at 850hPa is not available. Two of our ensembles (CORDEX-Africa 0.44° and CORDEX-Africa 0.22°) come from the CORDEX family and are lacking this variable. Hence, based on the variables that are already available within both CORDEX and CMIP5, we used potential temperature at 850 hPa (theta850) as an alternative "proxy" variable that provides thermodynamical information. In order to ensure that theta850 could be used instead of zg850, we examined the relationship between theta850 and zg850 over the study region in ERA5, for each month of the rainy season (Oct-Mar), using the climatological mean monthly values for the period 1986-2005. The comparison is depicted below as a series of maps and scatterplots. Each point in the scatterplots represents a pixel in the ERA5 dataset.

More specifically, in Figure 1 the mean monthly zg850 values for the period 1986-2005 are shown. During October, over the south-eastern part of Angola there is a region of low pressures. Moving towards the core of the rainy season, the low-pressure system deepens, while there seems to be a very weak extension of low pressures towards the south. In Figure 2 the mean monthly theta850 values for the period 1986-2005 are shown. As it is depicted, during October there is an array of high theta values located over south-eastern Angola, coinciding with the region of low zg850 values. As stated in Munday and Washington, (2017), this is indicative of the dry convection processes that are at play during the beginning of the rainy season over the region. Moving towards DJF, the high theta850 values move southwards, indicating that in the core of the rainy season, convection over the greater Angola region is not thermally induced, but there is a rather dynamical large-scale driver. Through Figure 1 and Figure 2 we concluded that although theta850 is not a perfect proxy for zg850, it can be used to identify certain aspects of the Angola low pressure system, such as its strength and location during the rainy season.

[Figure]

Figure 2: Mean monthly geopotential height at 850 hPa in ERA5 for the period 1986-2005.

[Figure]

Figure 3: Mean monthly potential temperature at 850 hPa in ERA5 for the period 1986-2005.

In addition, in Figure 3 the scatterplots between zg850 (x-axis) and theta850 (y-axis) for each month of the rainy season for the period 1986-2005 are shown, over the whole southern Africa (land pixels only). The same plot, but with pixels only from the greater Angola region (14 °E to 25 °E and from 11 °S to 19 °S) is displayed in Figure 4. Although the relationship between the two variables is not perfectly linear, they display a considerable association, especially over the greater Angola region (Figure 4).

[Figure]

Figure 4: Geopotential height at 850 hPa (x-axis) plotted against Potential temperature at 850 hPa (y-axis). Values refer to climatological monthly means for the period 1986-2005. Each dot in the scatterplot represents a pixel of the ERA5 dataset over the whole southern Africa region 10 °E to 42 °E and from 10 °S to 35 °S.

[Figure]

Figure 5: Geopotential height at 850 hPa (x-axis) plotted against Potential temperature at 850 hPa (y-axis). Values refer to climatological monthly means for the period 1986-2005. Each dot in the scatterplot represents a pixel of the ERA5 dataset over the whole southern Africa region 14 ºE to 25 ºE and from 11 ºS to 19 ºS.

Concerning relative vorticity (ζ) as used in Howard et al., 2018 we had to investigate the following issues: In Howard et al., 2018, they identify Angola Low events by using daily relative vorticity at 800 hPa. Although u and v wind components are available at 800 hPa in CMIP5/6, they are not available in CORDEX simulations. More specifically, in CORDEX-Africa, u and v wind components are only available at 850, 500 and 200 hPa. Hence, we had to investigate if we could use the 850 hPa pressure level (instead of 800) and if we did so, should we apply the same ζ threshold? In Howard et al., 2018, Angola Low events are identified within the region 14 ºE to 25 ºE and from 11 ºS to 19 ºS for mean daily ζ values $< -4 \times 10^{-5}$ s$^{-1}$. An additional issue that we took into account, is that u and v wind components at 850 hPa were not available on a daily timestep in CMIP6, but only on a monthly timestep. Hence, for consistency reasons we had to work with monthly files in all ensembles (both CMIP, CORDEX) and in ERA5. Lastly, some files from the CORDEX-Africa ensembles did not have complete timeseries (from 1986-2005), so they were not included in the calculation of the ensemble mean that eventually were used for the calculation of monthly relative vorticity. For CORDEX-Africa 0.22º these files were:

*850_AFR-22_MOHC-HadGEM2-ES_historical_r1i1p1_ICTP-RegCM4-7_v0.nc

*850_AFR-22_MPI-M-MPI-ESM-MR_historical_r1i1p1_ICTP-RegCM4-7_v0.nc

*850_AFR-22_NCC-NorESM1-M_historical_r1i1p1_ICTP-RegCM4-7_v0.nc

With regards to the fact that u and v wind components were available only on a monthly timestep in CMIP6, we compared the daily and monthly relative vorticity values at 800 hPa in ERA5 for all the months of the rainy season (Oct-Mar). The histograms are displayed below in Figure 5, with the daily $\zeta$ values as in Howard et al., 2018 on the left panel and the monthly values on the right. The difference in the y-axis results from the fact that when $\zeta$ is calculated using a daily timestep, the histogram is drawn using 5.421.825 values, while when the $\zeta$ is calculated using monthly u and v values, it is drawn using 178.200 values (for the period 1986-2005). The histograms display only cyclonic vorticities. Green lines display the threshold set by Howard et al., 2018 ($\zeta$ values $< -4 \times 10^{-5}$ s$^{-1}$), while red values display the threshold set by Desbiolles et al., 2020 ($\zeta$ values $< -1.5 \times 10^{-5}$ s$^{-1}$). As it is shown, using the distribution of the monthly values has a much shorter tail and the Howard et al., 2018 threshold appears to be very strict, as a criterion for the identification of Angola Low events.

[Figure]

Figure 6: Histogram of relative vorticity for months Oct-Mar during 1986-2005 in ERA5 using daily u and v values (left) and using monthly u and v values (right). Pixels used are enclosed by the region from 14 ºE to 25 ºE and from 11 ºS to 19 ºS.

With regards to the question of whether the 850 pressure level can be used instead of 800 hPa as in Howard et al., 2018, we examine monthly relative vorticity in ERA5 in both pressure levels, within the region from 14 ºE to 25 ºE and from 11 ºS to 19 ºS. The results are displayed in Figure 6. Both distributions are very similar in shape, maxima and spread, although the distribution of $\zeta$ values at 800 hPa appear to have a shorter tail. On both panels, both the Howard et al., 2018 and Desbiolles et al., 2020 thresholds are indicated. We conclude that 850 pressure level can be used instead of 800 hPa.

[Figure]

Figure 7: Histogram of relative vorticity for months Oct-Mar during 1986-2005 in ERA5 using u and v values at 800 hPa (left) and using u and v values at 850 hPa (right). Pixels used are enclosed by the region from 14 ºE to 25 ºE and from 11 ºS to 19 ºS. For both histograms mean monthly u and v values are used.

Lastly, with regards to the question of what the optimal threshold for the identification of Angola Low events in all datasets would be, we investigate the statistical distribution of mean monthly cyclonic vorticities in all ensembles used, for the 850 hPa pressure level. The results are displayed in Figure 7. In all histograms the Howard et al., 2018 and Desbiolles et al., 2020 thresholds are drawn. As it is indicated, the Howard et al., 2018 threshold is too strict and for 3 out of 4 ensembles it does not even correspond to existing $\zeta$ values. We conclude that the threshold used in Desbiolles et al., 2020 ($\zeta$ values $< -1.5 \times 10^{-5}$ s$^{-1}$) is reasonable, considering the shape of the distributions examined. However, when the Desbiolles et al., 2020 threshold was applied to the data, it was also found that it was too strict, especially for CMIP5/6. Hence, we now use monthly relative vorticity in order to identify Angola Low events, by employing the $\zeta$ values $< -1 \times 10^{-5}$ s$^{-1}$ threshold.

[Figure]

Figure 8: Histogram of relative vorticity for months Oct-Mar during 1986-2005 at 850 hPa for CORDEX-Africa at 0.22° (upper left), for CORDEX-Africa 0.44° (upper right), for CMIP5 (lower left), and for CMIP6 (lower right). Pixels used are enclosed by the region from 14 °E to 25 °E and from 11 °S to 19 °S. For all histograms mean monthly u and v values are used.

**4th Comment:**

Page 9, section 3.3. Apart from the strength of the Angola Low, its position also plays an important role, which I suggest being included.

**RESPONSE**: Thank you. We now include mean monthly maps of relative vorticity (applying the $\zeta < -1 \times 10^{-5}$ s$^{-1}$ threshold for the identification of Angola Low events) (shaded) and the potential temperature at 850 hPa overlayed on them in the form of contours.

**5th Comment:**

Page 10, Section 3.5. It would be good to also see how many models agree on the sign of the trends in addition to the significance in Fig S5.

**RESPONSE**: Thank you for this suggestion. We now include the following figure in the supplementary material, displaying the number of models in each ensemble that display either increasing or decreasing trends.

[Figure]

Figure 9: Number of ensemble members in each ensemble displaying increasing or decreasing trends.

---

## Author Response (AR2)

**Responses to Topical Editor**

**Comments to the author::**

"Dear Authors,

You have addressed very well the concerns of reviewer in your "Author's Response", but keeping only a fraction of these discussions in the revised manuscript. Even if the discussion remains available in the GMDD website, you should consider including more of those elements in the final revised paper in order to benefit to all readers.

rgds"

**AUTHOR'S RESPONSE**: We thank the Topical Editor for evaluating positively the responses we provided in the review process. Following the recommendation made for including more of the responses to the final revised paper, we now present a new revised manuscript. Under each respective response provided for each comment below, we now
provide in the category "**MINOR REVISIONS**" what further additions were made to the manuscript, concerning each respective comment. All new elements with regards to the new additions are given in blue, in the section "**MINOR REVISIONS**".

**Major comments:**

**1st Comment:**

I recommend more clearly laying out what the novelty or newness is of this work. Based on previous work it seems that precipitation in this region has been studied in similar ways before. Why is the method/results/approach in this work an improvement on those studies?

**AUTHOR'S RESPONSE**: Thank you very much for this comment. Indeed, precipitation over southern Africa has been studied before. More specifically, Nikulin et al. (2012) was the first to present an overview of the CORDEX-Africa ensemble and to analyze the spatiotemporal patterns of precipitation. They showed that during the rainy season (Jan-Mar as used in Nikulin et al. (2012)) there is a weak wet bias over southern Africa, and that the use of the ensemble mean was able to outperform individual models, highlighting the importance of ensemble-based
approaches. The Nikulin et al. (2012) analysis was conducted on a pan-African scale. Similarly, Kalognomou et al., (2013) analyzed the same ensemble of CORDEX-Africa simulations, focusing over southern Africa and reported similar findings to Nikulin et al., (2012). In Shongwe et al. (2014) a particular emphasis was put on the onset and retreat of the rainy season, especially over the eastern part of southern Africa. Nonetheless, as stated in Shongwe et al. (2014) "No attempt is made in this paper to identify the model physics and dynamics responsible
for the differences in RCM performance." All the aforementioned studies employed the evaluation (hindcast) simulations performed within CORDEX-Africa, driven by ERA-Interim; the analyzed ensemble was comprised of 10 RCMs. It is also worth mentioning that the regional climate model (RCM) versions used in the studies listed above, refer to previous versions of the respective RCMs, which have now been replaced by newer versions in more recent studies.

In Meque and Abiodun (2015) the same ensemble of 10 hindcast simulations was again used, but it was also compared with a set of CMIP5 GCM simulations, with the purpose to identify a causal association between ENSO and drought events over southern Africa. In Meque and Abiodun (2015) it was stated for the first time that RCMs were able to provide added value, compared to their driving GCMs. The issue of the added value of the CORDEX-Africa ensemble was clearly stated in Dosio et al. (2015), where 1 RCM participating in CORDEX-Africa (CCLM) was compared against 4 different driving GCMs. In Favre et al. (2016) a special focus was given on the annual cycle of precipitation over South Africa, using the same ensemble of 10 CORDEX-Africa hindcast simulations and in Abba Omar and Abiodun (2017), although the same hindcast ensemble was used, there was an effort to associate extreme precipitation events with dynamical processes such as the Tropical Temperate Troughs. A comprehensive assessment of the added value between historical CORDEX-Africa RCMs simulations and of their driving CMIP5 GCMs on a seasonal timescale over the whole of Africa, was performed in Dosio et al. (2019). The first time the CORDEX-Africa ensemble over southern Africa was compared with a plethora of observational and satellite products was presented in Abiodun et al., (2020), while the first time that CORDEX Africa at 0.44º and at 0.22º was analyzed compared to both CMIP5 and CMIP6 ensembles is presented in Dosio et al. (2021). More specifically, in Dosio et al. (2021) the analysis is performed on a seasonal timestep and on pan-African scale and its particular emphasis is placed on the projected changes of future precipitation, although a part of the analysis is dedicated to the period 1981-2010.

Our work aims to provide a comprehensive overview of the observed precipitation climatology particularly focusing over southern Africa, in all tools that are currently available in the climate community. For this reason, we employ all four ensembles used in Dosio et al. (2021) for the period 1986-2005 and we additionally employ a set of 12 observational (satellite, gridded and reanalysis) products. By doing so, we aim to highlight the precipitation uncertainty that exists even among different observational products, which is inherent in the methods used for their production. In addition, we attempt to make a connection between monthly precipitation climatology over southern Africa and a particularly important atmospheric feature, the Angola Low pressure system. To our knowledge, the Angola Low pressure system has not been studied yet within the context of CORDEX-Africa simulations. Although there has been an ample work of evaluating CORDEX-Africa simulations, we think that in order to better understand the reasons why RCM simulations do, or do not, display an improvement relative to their driving GCMs, there must be a shift towards process-based evaluations that examine particular (thermo)dynamic atmospheric processes over specific regions and specific time periods. For this reason, we also chose to perform our analysis on a monthly timescale during the rainy season (Oct-Mar). Often, seasonal means are conveniently used, however, seasonal averages might obscure spatio-temporal patterns that can only be identified on a finer temporal resolution. One of the main hindrances that often limits the ability to perform dynamic analysis in CORDEX-Africa (and CORDEX in general) simulations is the lack of available variables at different pressure levels. This was a shortcoming in our analysis also. In addition, we discuss the results with respect to monthly precipitation trends, as seen in all observational and modeling ensembles we use.

We agree that we need to present more clearly in the manuscript what the novelty and the newness of this work is. For this reason, we made the following changes in the Introduction:

**AUTHOR'S CHANGES IN MANUSCRIPT**: Section to last paragraph: "Therefore, in this paper we expand on previous research to investigate how monthly precipitation during the rainy season over southern Africa is simulated by different modelling systems, by analyzing the monthly precipitation climatologies, the interannual variability, specific precipitation indices and monthly precipitation trends during the period 1986-2005, in four different modeling systems (CORDEX 0.22º/0.44º, CMIP5/6) and observational ensembles (satellite, reanalysis and gridded datasets). Our main goal is to provide a comprehensive overview with regards to precipitation climatology over SAF as simulated by the state-of-the-art tools used by climate scientists. In addition, we investigate whether higher resolution models are able to provide an improved representation of precipitation over southern Africa and we investigated how a particularly important atmospheric feature, the Angola Low pressure system, is simulated in the RCM and GCM ensembles."

**MINOR REVISIONS:** In the Introduction we have added the following paragraph (now 3ʳᵈ paragraph), which summarizes the main efforts conducted within CORDEX-Africa, with regards to the evaluation of precipitation:

[revised manuscript text omitted]

Shongwe, M.E., Lennard, C., Liebmann, B., Kalognomou, E.-A., Ntsangwane, L., Pinto, I., 2014. An evaluation of
CORDEX regional climate models in simulating precipitation over Southern Africa. Atmospheric Sci. Lett. 16, 199–207. https://doi.org/10.1002/asl2.538

Wyser, K., van Noije, T., Yang, S., von Hardenberg, J., O'Donnell, D., Döscher, R., 2020. On the increased climate sensitivity in the EC-Earth model from CMIP5 to CMIP6. Geosci. Model Dev. 13, 3465–3474. https://doi.org/10.5194/gmd-13-3465-2020

**2nd Comment:**

In the second to last paragraph of the introduction the purpose and goals of the paper are given but there are several different statements of goals which I find somewhat unfocused. Is there a main goal that can be defined? It seems that the main focus of the paper is on how the RCM ensemble can be shown to be more useful for precipitation projections over this region compared to the GCMs but this is not clear. From the abstract it is also not very clear what are the main results the reader should see.

**AUTHOR'S RESPONSE**: Thank you very much for this comment and correction. Our analysis has two main goals: The
first goal is to provide an intercomparison of how monthly precipitation during the rainy season over southern Africa is simulated by different modelling systems (CORDEX 0.22°/0. 44°, CMIP5/6) and to also provide an overview of the spread that is seen even among the so called "observational products", highlighting the need for improved modeling and monitoring efforts over the region. Our second goal is indeed, to highlight how RCMs are able to address certain deficiencies identified in GCMs. The second to last paragraph of the introduction has now
changed to the following:

**AUTHOR'S CHANGES IN MANUSCRIPT**: "Therefore, in this paper we expand on previous research to investigate how monthly precipitation during the rainy season over southern Africa is simulated by different modelling systems, by analyzing the monthly precipitation climatologies, the interannual variability, specific precipitation indices and monthly precipitation trends during the period 1986-2005, in four different modeling systems (CORDEX
0.22°/0.44°, CMIP5/6) and observational ensembles (satellite, reanalysis and gridded datasets). Our main goal is to provide a comprehensive overview with regards to precipitation climatology over SAF as simulated by the state-of-the-art tools used by climate scientists. In addition, we investigate whether higher resolution models are able to provide an improved representation of precipitation over southern Africa and we investigated how a particularly important atmospheric feature, the Angola Low pressure system, is simulated in the RCM and GCM ensembles."

**3rd Comment:**

My understanding is that the CORDEX-Africa 0.22° data are available. If so, why was the older 50km dataset used when a newer one was available?

**AUTHOR'S RESPONSE**: Thank you for this comment. We have now included all the CORDEX-Africa 0.22° simulations available in the analysis. The CORDEX-Africa 0.22° simulations added, are listed in the table below. We have kept, however, all CORDEX-Africa 0.44° simulations, since they constitute a larger ensemble (26 ensemble members used).

| Driving GCMs | RCMs | Realisations | Variables |
|---|---|---|---|
| CanESM2 | CanRCM4 | r1i1p1 | Pr, |
| HadGEM2-ES | CCLM5-0-15 | r1i1p1 | Pr, hus850, ua850, va850, ta850 |
| | REMO2015 | r1i1p1 | Pr, hus850, ua850, va850, ta850 |
| | RegCM4-7 | r1i1p1 | Pr, hus850, ua850, va850, ta850 |
| MPI-ESM-LR | CCLM5-0-15 | r1i1p1 | Pr, hus850, ua850, va850, ta850 |
| | REMO2015 | r1i1p1 | Pr, hus850, ua850, va850, ta850 |
| | RegCM4-7 | r1i1p1 | Pr, hus850, ua850, va850, ta850 |
| NorESM1-M | CCLM5-0-15 | r1i1p1 | Pr, hus850, ua850, va850, ta850 |
| | REMO2015 | r1i1p1 | Pr, hus850, ua850, va850, ta850 |
| | RegCM4-7 | r1i1p1 | Pr, hus850, ua850, va850, ta850 |

**Minor Comment:**

**1st Comment:**

Line 17: SAF hasn't been defined yet, it should be defined here.

**AUTHOR'S RESPONSE**: We have defined SAF in line 10 (beginning of the abstract).

**2nd Comment:**

Lines 22-23: "…a similar behavior to CMIP5, however reducing slightly the ensemble spread." I would replace 'however' here with 'but'.

**AUTHOR'S RESPONSE**: Thank you! We made this change in the manuscript.

**AUTHOR'S CHANGES IN MANUSCRIPT**: The sentence now reads: "The CMIP6 ensemble displayed a similar behaviour to CMIP5 but reducing slightly the ensemble spread."

**3ʳᵈ Comment:**

Line 61: Over what period is this trend seen? I assume it's a historical period but it would be good to explicitly say it here.

**AUTHOR'S RESPONSE**: Thank you! We will made this clarification in the manuscript.

**AUTHOR'S CHANGES IN MANUSCRIPT**: "During DJF, precipitation trends during the historical over SAF display…".

**4ᵗʰ Comment:**

Sentence starting at Line 71 "However,…": This sentence is a little bit confusing I would recommend removing 'still' and the comma between 'period' and 'persist'.

**AUTHOR'S RESPONSE**: Thank you! The sentence in the manuscript was changed to:

**AUTHOR'S CHANGES IN MANUSCRIPT**: "However, although the CMIP6 ensemble exhibits multiple improvements on various levels (Wyser et al., 2020), certain biases and challenges identified in CMIP5 during the historical period persist in CMIP6 (Kim et al., 2020)."

      **5ᵗʰ Comment:**

Line 90: Provide more detail of what will be addressed in the results section (Section 3). For instance, describe the subsections of the results and what will be covered.

**AUTHOR'S RESPONSE**: Thank you very much for this comment! This sentence in the manuscript was changed to:

**AUTHOR'S CHANGES IN MANUSCRIPT**: "In Section 3 the results are presented. More specifically, the results are analyzed based on the monthly climatology, the annual cycle of precipitation, the Angola Low pressure system,
the ETCCDI precipitation indices and the monthly precipitation trends. Lastly, in Section 4 we provide the discussion of the analysis along with some concluding remarks."

**6ᵗʰ Comment:**

Line 107: Should this be "less **than** or equal to"?

**AUTHOR'S RESPONSE**: Thank you very much.

     **AUTHOR'S CHANGES IN MANUSCRIPT**: We changed the sentence to "The gauge-based products were chosen so that they have a spatial resolution less than or equal to 0.5° x 0.5°…"

**7th Comment:**

Line 183: How was the calculation of standard deviation done to get the within-ensemble agreement? Was the monthly mean of over the 1986-2005 period calculated for each model first and then the SD of the ensemble taken?

     **AUTHOR'S RESPONSE**: Yes, we first calculated the monthly means over the period 1986-2005 for each model (or observational dataset) separately, and then we calculate the standard deviation.

**8th Comment:**

Figures 1, 2 and 7: The alignment and spacing of the panels is not consistent. I recommend making sure the Figures have consistent spacing and are aligned to improve their visual aesthetic.

     **AUTHOR'S RESPONSE**: Thank you! We made this correction in all panel plots.

**9th Comment:**

Lines 355-356: Expand on what improvements can be made. This is an important statement for readers who may be
interested in expanding on this work.

     **AUTHOR'S RESPONSE:** Thank you very much for this comment. The paragraph in the manuscript has been changed to the following:

**AUTHOR'S CHANGES IN MANUSCRIPT:** "In conclusion, while CORDEX0.44 displays marked improvement over coarser resolution products, there are still further improvements to be made. More specifically, since the wet bias in RCM simulations persists (although considerably reduced relative to GCMs), it is necessary that precipitation over southern Africa is no longer assessed based on bulk descriptive statistics, but that there will be a shift towards process-based evaluation, where the dynamical and thermodynamical characteristics of specific atmospheric
features are investigated more thoroughly in the CORDEX-Africa simulations. For this reason, it is imperative that all institutes submitting RCM simulations in data repositories such as the Earth System Grid Federation or the Copernicus Climate Data Store, provide model output data on multiple pressure levels, so that a fair comparison with the CMIP community would be possible. In addition, since the climate of southern Africa is highly coupled with the moisture transport coming from the adjacent oceans, it is necessary that the next generation of RCM
simulations within CORDEX-Africa are performed coupled with ocean models. Lastly, since convection over southern Africa has a strong thermal component during specific months of the year (Oct-Nov), it is necessary that the land-atmosphere coupling processes within each RCM are examined in more detail, with coordinated efforts such as the LUCAS Flagship Pilot Study (https://ms.hereon.de/cordex_fps_lucas/index.php.en), as performed in the Euro-CORDEX domain. In the world of regional climate modelling community, the 0.44° resolution of
CORDEX-Africa is no longer state of the art and ensemble efforts are now approaching convection permitting grid-spacing (i.e., < 4 km) in some parts of the world (Ban et al., 2021; Pichelli et al., 2021) (Ban et al., 2021;

Pichelli et al., 2021). We also note, that increasing effort should be made with regards to understanding the improvements made from CORDEX0.44 simulations to CORDEX0.22. Although higher resolution is a desired target in the climate modelling community due to the more realistic representation of processes that it offers, still it
should not be used as a panacea. In the current work we identified certain weaknesses in the CORDEX0.22 ensemble, that should be addressed before the community populates further its simulation matrix. The next generation ensembles for Africa will hopefully provide insight and improvements to the challenges described here."

**Responses to Anonymous Reviewer 2**

**General Comment:**

This paper evaluates the representation of the southern African rainfall in the GCMs and RCMs compared to a set of observational data. The rainfall climatology, annual cycle, trends and a couple of ETCCDI indices are analyzed along with the representation of the Angola Low, which is one of the important driving circulations that affect the rainfall in the area. The paper is of high importance for model improvement. However, I suggest the following comments to be addressed before the paper is published in GMD.

**AUTHOR'S RESPONSE**: We would like to thank the Anonymous Reviewer #2 for the positive interpretation of the manuscript. Based on the suggestions and comments, we provide the following replies.

**Major comments:**

**1st Comment:**

Page 8, 235-240, an evaluation of the moisture transported through the north-easterly monsoon should be performed here to support the hypothesis that the improved representation of the topography led to a lower bias in the CORDEX models.

**AUTHOR'S RESPONSE**: Thank you very much for this comment. We now include the following figure in the main manuscript, displaying the moisture flux and moisture flux divergence at 850 hPa during each month of the rainy
season, for the period 1986-2005. More specifically, the moisture flux divergence was calculated using the product of specific humidity and wind at 850 hPa, following the equation below (the vertical component ($\frac{\partial qw}{\partial z}$) is considered negligible).

$$\nabla \cdot q\vec{u} = \frac{\partial qu}{\partial x} + \frac{\partial qv}{\partial y}$$

With this plot we aim to contribute to the discussion developed in Figure 11 in Munday and Washington, (2017). More specifically, one of the reasons responsible for the wet bias of CMIP5 models over southern Africa (SAF), was that mountainous regions over the northeast part of SAF were underrepresented, due to the spatial resolution of the CMIP5 models Munday and Washington, (2018). The high elevation areas over Malawi and Tanzania were not represented accurately in CMIP5 GCMs, which allowed moisture transport entering SAF from the northeast to penetrate central SAF, rather than to recurve around the high mountains and result to large precipitation amounts over northern Madagascar. Since the underrepresentation of topography in GCMs is a matter of spatial resolution, we make the hypothesis that in high resolution RCMs this issue is resolved, since moisture entering SAF from the northeast is blocked by the adequately high elevation over the Tanzania and Malawi region.

As seen in the Figure 1 below, during all months the moisture flux field is very spatially inhomogeneous in ERA5 and in both CORDEX ensembles, while in CMIP5/6 the field is considerably smoother, indicating that in low resolution GCMs the surface characteristics are not detailed enough, so as to allow for adequate friction and cause the moisture fluxes to recurve around mountainous areas. Particularly during December and January when the north-easterly monsoon is intensified, the moisture flux at the northeast of SAF is intercepted in both CORDEX ensembles, however not in CMIP5/6. After February the atmospheric flow from the northeast is weakened and it is strengthened at the southeastern part, entering SAF through Mozambique. This moisture transport originates from the Mascarene High that has developed over the South Indian Ocean. The recurvature of moisture seen at the south-eastern part of Mozambique is caused by the Mozambique Channel Trough (Barimala et al., 2018).

**AUTHOR'S CHANGES IN MANUSCRIPT**: In the manuscript we comment concerning the moisture transport entering SAF from the northeastern part, by adding the following text as the last sentence of paragraph 3 in Section 3.2: "The improvement of orography has a further effect in blocking moisture transport entering SAF from the northeast, especially during Dec-Jan, as seen in Fig. 5."

[Figure]

Figure 1: Moisture flux and divergence at 850 hPa.

**2ⁿᵈ Comment:**

Page 9, section 3.3. It should be made clear why there is a special focus on the Angola low given the different processes that significantly affect the rainfall in the area. For example, the cloudband or tropical temperate trough is one of the major processes that drive rainfall in SAF but is never mentioned here. I would even suggest including the cloudbands in the analyzes.

**AUTHOR'S RESPONSE**: Thank you very much for this comment. Indeed, not mentioning the Tropical Temperate Troughs (TTTs) in the manuscript is a lack, since TTTs are one of the main mechanisms producing precipitation over southern Africa. We now refer to the role they play for precipitation over southern Africa in the results section (section 3.3), where findings about the Angola Low have further implications for the formation of TTTs.

More specifically, the reason why we chose to put an emphasis on the Angola Low pressure system is that usually Angola Low events precede the formation of TTTs and hence, they can be considered as their precursor in the "climate process chain" controlling precipitation over southern Africa (Daron et al., 2019). As stated in Howard et al., 2018, it is common that Angola Low events precede TTT events, since the Angola Low pressure system functions as a key process necessary for the transport of water vapor from the tropics towards the extratropics (Hart et al., 2010).

In addition, based on a Scopus query investigating the number of documents with the keywords "Angola Low" and "Tropical Temperate Troughs" in the Title-Abstract-Keywords, we saw that TTTs have received almost the double attention in the literature (47 published papers), relative to the Angola Low (23 published papers). Hence, our work is, in part, an attempt to address this gap, considering the limitations set by the availability of variables in all the ensembles that are currently examined (CORDEX-Africa 0.22º/0.44º and CMIP5/6). For this reason, we did not include an analysis of the TTTs, since it is beyond of the scope of the current study, but it is imperative that a comparative analysis of how TTTs are simulated in CORDEX-Africa 0.22º/0.44º and CMIP5/6 is performed.

**MINOR REVISIONS**: In lines **749-755** we now include the following sentences:

"More specifically, the reason why we chose to put an emphasis on the AL pressure system, is that the AL redistributes low-tropospheric moisture entering SAF from the southern Atlantic and the southern Indian oceans and also, moisture transport originating from the Congo basin. In addition, AL events precede the formation of Tropical Temperate Troughs (TTTs) and hence, they can be considered as their precursor in the "climate process chain (Daron et al., 2019). As stated in Howard and Washington (2018), it is common that AL events precede TTT events, since the AL pressure system functions as a key process necessary for the transport of water vapor from the tropics towards the extratropics (Hart et al., 2010)."

**3rd Comment:**

Page 9, section 3.3. I wonder why theta850 is used to calculate the Angola low instead of the geopotential height (as in Munday et al., 2017) or the vorticity (as in Howard et al., 2018). The CMIP6 models do have these variables available and should be used for a fair comparison.

**AUTHOR'S RESPONSE**: Indeed, Munday and Washington, (2017) use the lowest 5% of mean DJF geopotential height at 850 hPa (zg850) over southern Africa. The reason why we were not able to use the same index in order to identify the Angola Low, was that within the context of CORDEX-Africa simulations, geopotential height at 850hPa is not available. Two of our ensembles (CORDEX-Africa 0.44º and CORDEX-Africa 0.22º) come from the CORDEX family and are lacking this variable. Hence, based on the variables that are already available within both CORDEX and CMIP5, we used potential temperature at 850 hPa (theta850) as an alternative "proxy" variable that provides thermodynamical information. In order to ensure that theta850 could be used instead of zg850, we examined the relationship between theta850 and zg850 over the study region in ERA5, for each month of the rainy season (Oct-Mar), using the climatological mean monthly values for the period 1986-2005. The comparison is depicted below as a series of maps and scatterplots. Each point in the scatterplots represents a pixel in the ERA5 dataset.

More specifically, in Figure 1 the mean monthly zg850 values for the period 1986-2005 are shown. During October, over the south-eastern part of Angola there is a region of low pressures. Moving towards the core of the rainy season, the low-pressure system deepens, while there seems to be a very weak extension of low pressures towards the south. In Figure 2 the mean monthly theta850 values for the period 1986-2005 are shown. As it is depicted, during October there is an array of high theta values located over south-eastern Angola, coinciding with the region of low zg850 values. As stated in Munday and Washington, (2017), this is indicative of the dry convection processes that are at play during the beginning of the rainy season over the region. Moving towards

DJF, the high theta850 values move southwards, indicating that in the core of the rainy season, convection over the greater Angola region is not thermally induced, but there is a rather dynamical large-scale driver. Through Figure 1 and Figure 2 we concluded that although theta850 is not a perfect proxy for zg850, it can be used to identify
certain aspects of the Angola low pressure system, such as its strength and location during the rainy season.

[Figure]

Figure 2: Mean monthly geopotential height at 850 hPa in ERA5 for the period 1986-2005.

[Figure]

Figure 3: Mean monthly potential temperature at 850 hPa in ERA5 for the period 1986-2005.

In addition, in Figure 3 the scatterplots between zg850 (x-axis) and theta850 (y-axis) for each month of the rainy season for the period 1986-2005 are shown, over the whole southern Africa (land pixels only). The same plot, but with pixels only from the greater Angola region (14 ºE to 25 ºE and from 11 ºS to 19 ºS) is displayed in Figure 4. Although the relationship between the two variables is not perfectly linear, they display a considerable association, especially over the greater Angola region (Figure 4).

[Figure]

Figure 4: Geopotential height at 850 hPa (x-axis) plotted against Potential temperature at 850 hPa (y-axis). Values refer to climatological monthly means for the period 1986-2005. Each dot in the scatterplot represents a pixel of the ERA5 dataset over the whole southern Africa region 10 °E to 42 °E and from 10 °S to 35 °S.

[Figure]

Figure 5: Geopotential height at 850 hPa (x-axis) plotted against Potential temperature at 850 hPa (y-axis). Values refer to climatological monthly means for the period 1986-2005. Each dot in the scatterplot represents a pixel of the ERA5 dataset over the whole southern Africa region 14 °E to 25 °E and from 11 °S to 19 °S.

Concerning relative vorticity ($\zeta$) as used in Howard et al., 2018 we had to investigate the following issues: In Howard et al., 2018, they identify Angola Low events by using daily relative vorticity at 800 hPa. Although u and v wind components are available at 800 hPa in CMIP5/6, they are not available in CORDEX simulations. More specifically, in CORDEX-Africa, u and v wind components are only available at 850, 500 and 200 hPa. Hence, we had to investigate if we could use the 850 hPa pressure level (instead of 800) and if we did so, should we apply the same $\zeta$ threshold? In Howard et al., 2018, Angola Low events are identified within the region 14 °E to 25 °E and

from 11 °S to 19 °S for mean daily $\zeta$ values $< -4 \times 10^{-5}$ s$^{-1}$. An additional issue that we took into account, is that u and v wind components at 850 hPa were not available on a daily timestep in CMIP6, but only on a monthly timestep. Hence, for consistency reasons we had to work with monthly files in all ensembles (both CMIP, CORDEX) and in ERA5. Lastly, some files from the CORDEX-Africa ensembles did not have complete timeseries (from 1986-2005), so they were not included in the calculation of the ensemble mean that eventually were used for

the calculation of monthly relative vorticity. For CORDEX-Africa 0.22° these files were:

*850_AFR-22_MOHC-HadGEM2-ES_historical_r1i1p1_ICTP-RegCM4-7_v0.nc

*850_AFR-22_MPI-M-MPI-ESM-MR_historical_r1i1p1_ICTP-RegCM4-7_v0.nc

*850_AFR-22_NCC-NorESM1-M_historical_r1i1p1_ICTP-RegCM4-7_v0.nc

With regards to the fact that u and v wind components were available only on a monthly timestep in CMIP6, we compared the daily and monthly relative vorticity values at 800 hPa in ERA5 for all the months of the rainy season (Oct-Mar). The histograms are displayed below in Figure 6, with the daily $\zeta$ values as in Howard et al., 2018 on the left panel and the monthly values on the right. The difference in the y-axis results from the fact that when $\zeta$ is calculated using a daily timestep, the histogram is drawn using 5.421.825 values, while when the $\zeta$ is calculated using monthly u and v values, it is drawn using 178.200 values (for the period 1986-2005). The histograms display only cyclonic vorticities. Green lines display the threshold set by Howard et al., 2018 ($\zeta$ values $< -4 \times 10^{-5}$ s$^{-1}$), while red values display the threshold set by Desbiolles et al., 2020 ($\zeta$ values $< -1.5 \times 10^{-5}$ s$^{-1}$). As it is shown, using the distribution of the monthly values has a much shorter tail and the Howard et al., 2018 threshold appears to be very strict, as a criterion for the identification of Angola Low events.

[Figure]

Figure 6: Histogram of relative vorticity for months Oct-Mar during 1986-2005 in ERA5 using daily u and v values (left) and using monthly u and v values (right). Pixels used are enclosed by the region from 14 ºE to 25 ºE and from 11 ºS to 19 ºS.

With regards to the question of whether the 850 pressure level can be used instead of 800 hPa as in Howard et al., 2018, we examine monthly relative vorticity in ERA5 in both pressure levels, within the region from 14 ºE to 25 ºE and from 11 ºS to 19 ºS. The results are displayed in Figure 6. Both distributions are very similar in shape, maxima and spread, although the distribution of $\zeta$ values at 800 hPa appear to have a shorter tail. On both panels, both the Howard et al., 2018 and Desbiolles et al., 2020 thresholds are indicated. We conclude that 850 pressure level can be used instead of 800 hPa.

[Figure]

Figure 7: Histogram of relative vorticity for months Oct-Mar during 1986-2005 in ERA5 using u and v values at 800 hPa (left) and using u and v values at 850 hPa (right). Pixels used are enclosed by the region from 14 °E to 25 °E and from 11 °S to 19 °S. For both histograms mean monthly u and v values are used.

Lastly, with regards to the question of what the optimal threshold for the identification of Angola Low events in all datasets would be, we investigate the statistical distribution of mean monthly cyclonic vorticities in all ensembles used, for the 850 hPa pressure level. The results are displayed in Figure 7. In all histograms the Howard et al., 2018 and Desbiolles et al., 2020 thresholds are drawn. As it is indicated, the Howard et al., 2018 threshold is too strict and for 3 out of 4 ensembles it does not even correspond to existing $\zeta$ values. We conclude that the threshold used in Desbiolles et al., 2020 ($\zeta$ values $< -1.5 \times 10^{-5}$ s$^{-1}$) is reasonable, considering the shape of the distributions examined. However, when the Desbiolles et al., 2020 threshold was applied to the data, it was also found that it was too strict, especially for CMIP5/6. Hence, we now use monthly relative vorticity in order to identify Angola Low events, by employing the $\zeta$ values $< -1 \times 10^{-5}$ s$^{-1}$ threshold.

[Figure]

[Figure]

Figure 8: Histogram of relative vorticity for months Oct-Mar during 1986-2005 at 850 hPa for CORDEX-Africa at 0.22º (upper left), for CORDEX-Africa 0.44º (upper right), for CMIP5 (lower left), and for CMIP6 (lower right). Pixels used are enclosed by the region from 14 ºE to 25 ºE and from 11 ºS to 19 ºS. For all histograms mean monthly u and v values are used.

**MINOR REVISIONS**: In lines **756-797** we have added the following section, which is mainly taken from the answer
provided above.

[revised manuscript text omitted]

**4th Comment:**

Page 9, section 3.3. Apart from the strength of the Angola Low, its position also plays an important role, which I suggest being included.

**AUTHOR'S RESPONSE**: Thank you. We now include mean monthly maps of relative vorticity (applying the $\zeta < -1 \times 10^{-5}$ s$^{-1}$ threshold for the identification of Angola Low events) (shaded) and the potential temperature at 850 hPa overlayed on them in the form of contours.

**AUTHOR'S CHANGES IN MANUSCRIPT:** The following figure has been added displaying monthly climatologies of the Angola Low pressure system during the rainy season for the period 1986-2005. Filled contours indicate cyclonic relative vorticity ($\zeta$) for $\zeta < -0.00001$ s$^{-1}$ over the region extending from 14 °E to 25 °E and from 11 °S to 19 °S. Red lines indicate the isotherms of potential temperature at 850 hPa, having an increment of 2 K. Blue lines indicate isoheights of the geopotential height at 850 hPa, having an increment of 5 m. CORDEX0.44/0.22 are not plotted with geopotential isoheights, because this variable was not available for CORDEX simulations. From top to bottom: ERA5, ensemble mean of CORDEX0.22°, CORDEX0.44°, CMIP5 and CMIP6 simulations.

[Figure]

**5th Comment:**

Page 10, Section 3.5. It would be good to also see how many models agree on the sign of the trends in addition to the significance in Fig S5.

**AUTHOR'S RESPONSE**: Thank you for this suggestion. We now include the following figure in the supplementary material, displaying the number of models in each ensemble that display either increasing or decreasing trends.

[Figure]

Figure 9: Number of ensemble members in each ensemble displaying increasing or decreasing trends.